# Leiomodin creates a leaky cap at the pointed end of actin-thin filaments

**Dmitri Tolkatchev**[ID][1]☯*, **Garry E. Smith, Jr.**[ID][1]☯, **Lauren E. Schultz**[ID][2], **Mert Colpan**[ID][2], **Gregory L. Helms**[3], **John R. Cort**[ID][4,5], **Carol C. Gregorio**[2]*, **Alla S. Kostyukova**[ID][1]*

**1** Voiland School of Chemical Engineering and Bioengineering, Washington State University, Pullman, Washington, United States of America, **2** Department of Cellular and Molecular Medicine and Sarver Molecular Cardiovascular Research Program, The University of Arizona, Tucson, Arizona, United States of America, **3** The Center for NMR Spectroscopy, Washington State University, Pullman, Washington, United States of America, **4** Earth and Biological Sciences Directorate, Pacific Northwest National Laboratory, Richland, Washington, United States of America, **5** Institute of Biological Chemistry, Washington State University, Pullman, Washington, United States of America

☯ These authors contributed equally to this work.
* alla.kostyukova@wsu.edu (ASK); dmitri.tolkatchev@wsu.edu (DT); gregorio@email.arizona.edu (CGG)

**Data Availability Statement:** The foregoing restrictions apply to NMR chemical shifts submitted to the BMRB database, accession number 30681; and 3D coordinates of the complex submitted to the RCSB PDB, ID 6UT2. These data

## Abstract

Improper lengths of actin-thin filaments are associated with altered contractile activity and lethal myopathies. Leiomodin, a member of the tropomodulin family of proteins, is critical in thin filament assembly and maintenance; however, its role is under dispute. Using nuclear magnetic resonance data and molecular dynamics simulations, we generated the first atomic structural model of the binding interface between the tropomyosin-binding site of cardiac leiomodin and the N-terminus of striated muscle tropomyosin. Our structural data indicate that the leiomodin/tropomyosin complex only forms at the pointed end of thin filaments, where the tropomyosin N-terminus is not blocked by an adjacent tropomyosin protomer. This discovery provides evidence supporting the debated mechanism where leiomodin and tropomodulin regulate thin filament lengths by competing for thin filament binding. Data from experiments performed in cardiomyocytes provide additional support for the competition model; specifically, expression of a leiomodin mutant that is unable to interact with tropomyosin fails to displace tropomodulin at thin filament pointed ends and fails to elongate thin filaments. Together with previous structural and biochemical data, we now propose a molecular mechanism of actin polymerization at the pointed end in the presence of bound leiomodin. In the proposed model, the N-terminal actin-binding site of leiomodin can act as a "swinging gate" allowing limited actin polymerization, thus making leiomodin a leaky pointed-end cap. Results presented in this work answer long-standing questions about the role of leiomodin in thin filament length regulation and maintenance.

## Introduction

Actin filaments play essential roles in muscle contraction, cytokinesis, cell motility, and cell morphology. Filamentous actin (F-actin) is a polymer of globular actin (G-actin) subunits [1].

will become available once the paper is accepted for publication. All other relevant data are within the paper and its Supporting Information files.

**Funding:** The study was funded by the National Institutes of Health (NIH) grant GM120137 to ASK and CCG, https://www.nih.gov/; the NIH grant HL123078 to CCG, https://www.nih.gov/; the NIH/ National Institute of General Medical Sciences– funded protein biotechnology training program T32 GM008336 to GES, https://www.nigms.nih.gov/ training/instpredoc/pages/predocdesc- biotechnology.aspx; the American Heart Association Postdoctoral Fellowship 19POST34450023 to MC, https://professional. heart.org/professional/ResearchPrograms/ ApplicationInformation/UCM_443314_ Postdoctoral-Fellowship.jsp; the NHLBI training program T32HL007249 to LES, https://www.nhlbi. nih.gov/; the NIH grant 1F31AR076209 to LES, https://www.nih.gov/. The funders had no role in study design, data collection and analysis, decision to publish, or preparation of the manuscript.

**Competing interests:** The authors have declared that no competing interests exist.

**Abbreviations:** 3D, three-dimensional; ABS, actin- binding site; CD, circular dichroism; F-actin, filamentous actin; G-actin, globular actin; GFP, green fluorescent protein; HSQC, heteronuclear single-quantum coherence; Lmod, leiomodin; LRR, leucine-rich repeat; MDS, molecular dynamics simulation; NOE, nuclear Overhauser effect; PDB, Protein Data Bank; RCI, random coil index; RDC, residual dipolar coupling; RMSD, root-mean- square deviation; Tmod, tropomodulin; TpmBS, tropomyosin-binding site; WT, wild type.

The helical assembly of F-actin is polar, with its different ends termed barbed and pointed end. Another polymeric protein, tropomyosin, runs longitudinally along each of the two sides of F-actin [2]. Each tropomyosin molecule is composed of two α-helical polypeptide chains, assembled in a parallel coiled coil, with their N-termini oriented toward the pointed end and their C-termini toward the barbed end. Tropomyosin polymerizes into a cable along the length of the actin filament via head-to-tail (N-terminus to C-terminus) overlap with adjacent tropomyosin molecules.

In striated muscles, optimal length of actin-thin filaments is a critical factor for efficient contractile activity, and alterations in thin filament lengths are linked to devastating human skeletal and cardiac muscle diseases. Mutations interfering with the regulation of thin filament length are associated with nemaline myopathy [3, 4] and dilated cardiomyopathy [5, 6]. The mechanisms and important players required for thin filament length regulation are not fully identified (e.g., [7]). Currently, there is a debate on the true mechanism(s) associated with the regulatory roles of molecular interactions occurring at thin filament pointed ends (for reviews, see [8–10]).

The pointed end of thin filaments in striated muscle sarcomeres is located near the M-line, and a capping protein, tropomodulin (Tmod), blocks further actin filament polymerization and depolymerization [11]. The pointed-end binding of Tmod stabilizes thin filaments and has an important role in thin filament length regulation [12, 13]. Tmod interacts with the thin filament via four distinct, sequentially alternating tropomyosin- and actin-binding sites (TpmBSs and ABSs, respectively) [13–16], numbered according to their location in the Tmod amino acid sequence: TpmBS1, ABS1, TpmBS2, and ABS2. Two TpmBSs and one ABS (TpmBS1, ABS1, TpmBS2) are in the intrinsically disordered N-terminal half of Tmod [17], whereas the second actin-binding site represents the C-terminal half of the protein [16], which is a folded leucine-rich repeat (LRR) [18]. Tmod specificity for the pointed end is ensured by TpmBS1 and TpmBS2 binding simultaneously to two tropomyosin molecules on two opposite sides of the thin filament [14]. The binding is reinforced by the interactions of ABS1 and ABS2 with actin. Once Tmod is fully bound to the pointed end, it sterically hinders the attachment of actin monomers to the pointed end and consequently prevents further filament elongation [14, 19].

Leiomodin (Lmod) is a homolog of Tmod and a strong actin nucleator in biochemical assays [20]. Excess Lmod2 (the cardiac Lmod isoform) leads to elongation of thin filaments [21], and constitutive or conditional knockout of Lmod2 results in thin filament shortening [5, 22]. Hence, it was proposed that Lmod2 has a specific role in the maintenance of mature thin filament lengths. Indeed, both Lmod2 and Tmod1 (the major Tmod isoform present in cardiac muscle) are localized to the pointed end of thin filaments in cardiomyocytes [20, 21]. In addition, Lmod2 affects pointed-end polymerization of actin/tropomyosin filaments in biochemical assays and displaces Tmod1 both in biochemical assays and in cardiomyocytes [21, 23, 24]. Therefore, it has been suggested that Lmod2 and Tmod1 regulate thin filament length via dynamic exchange between the two molecules at the pointed end [21].

There are two contested models for how Lmod2 functions at the thin filament pointed end. In the proposed "competition model," Lmod2 and Tmod1 can displace each other from thin filament pointed ends. Lmod2, when bound to the pointed end, allows the thin filament to elongate, albeit slowly [21, 23]. However, when Tmod1 is bound instead, the pointed end is capped and elongation is prohibited. Then, the dynamic interactions of Tmod1 and Lmod2 with actin/tropomyosin at the pointed end are an essential factor contributing to the final length of the thin filaments.

The function of Lmod2 as an elongation factor, a key aspect of the competition model, is not universally accepted, and hence, it is a subject of debate [8–10]. According to an alternative

"nucleation model," Lmod functions solely as a nucleator of new thin filaments during sarcomere assembly in developing muscles and during sarcomere repair and turnover in mature muscles [20, 25]. Prior to this work, no atomic structure of a binding interface, which places Lmod2 specifically at the pointed end, has been known. It has been reasoned that Lmod2 does not compete with Tmod1 for the pointed end, and its only function, by elimination, is nucleation [9].

Key to the validity of the competition model is the dynamic exchange between Lmod2 and Tmod1, which is only possible if they compete for binding to the pointed end and therefore can displace each other in vivo. Similarly to its homolog Tmod1, Lmod2 contains an intrinsically disordered N-terminal region with an affinity for both tropomyosin and actin [23, 26–28] and an actin-binding LRR domain [20, 25]. It has been previously determined that TpmBS1 of Lmod2 (residues Arg7-Glu41) is very similar to that of Tmod1 (residues Ser2-Asp38), with approximately 50% identity and approximately 75% similarity between them [10, 27, 29]. The three-dimensional (3D) structural information on the binding of tropomyosin with either Lmod or Tmod is lacking. However, on the basis of the close sequence resemblance, it can be assumed that the mode of interaction of Tmod or Lmod TpmBS1 with tropomyosin is structurally homologous. It is not known, however, if they bind to the same tropomyosin protomer on the thin filament. Since the hypothetical models of the complexes of TpmBS1 and tropomyosin do not agree [15, 19], in this study we determined the 3D structure of the Lmod2/tropomyosin complex. The structure elucidates the true mechanism of action of Lmod and Tmod with respect to thin filament length regulation.

Here, we present NMR studies and NMR-guided molecular dynamics simulations (MDSs) that resulted in a novel 3D model of a complex between peptides representing Lmod2 TpmBS1 and the tropomyosin Lmod2-binding site. The model was validated by experiments with modified tropomyosin peptides, which produced NMR spectral effects consistent with the 3D model. Another critical validation came from $^1$H-$^{15}$N residual dipolar coupling (RDC) values, which provide the orientations of internuclear vectors [30]. The model gives novel insights into the mode of tropomyosin interaction with members of the Tmod family. Specifically, our structural data indicate that binding of Tmod and Lmod (via TpmBS1) to tropomyosin can only be realized at the pointed end, where the N-terminus of tropomyosin is available (i.e., binding can only happen at the terminal tropomyosin molecule that is free and not obstructed by another tropomyosin molecule). Moreover, we showed that cardiomyocyte expression of Lmod2 carrying a mutation designed to weaken Lmod2 interaction with tropomyosin resulted in disrupted subcellular assembly, impaired ability to displace Tmod1, and the inability to elongate thin filaments like wild-type (WT) Lmod2. We suggest a structure-based molecular mechanism for Lmod functioning as a leaky cap at the pointed end. The mechanism explains how actin can polymerize at the pointed end of the thin filament when Lmod is bound.

## Results and discussion

### Building the 3D structure of the Lmod2s1/αTM1a$_{1-14}$Zip complex

**The 3D structure of the Lmod2s1/αTM1a$_{1-14}$Zip complex represents the binding interface between Lmod2 and tropomyosin.**   Lmod2s1 and αTM1a$_{1-14}$Zip (amino acid sequences shown in S1 Fig) were chosen as peptides representing Lmod2 TpmBS1 and the tropomyosin Lmod2-binding site, respectively [27]. The αTM1a$_{1-14}$Zip peptide forms a stable coiled-coil dimer under most of the conditions used in this study, and we will henceforth use the name αTM1a$_{1-14}$Zip to refer to the dimeric molecule. The stoichiometric ratios are listed hereafter with respect to the molar amount of dimer.

The obtained model of the Lmod2s1/αTM1a$_{1-14}$Zip complex is shown in Fig 1. In the complex, two α-helices of Lmod2s1 bind to the N-terminal part of the coiled-coil αTM1a$_{1-14}$Zip, thus resulting in a 4-helix bundle. The two-residue loop Leu25-Ser26 connecting two α-helices of Lmod2s1 is positioned above the two N-termini of αTM1a$_{1-14}$Zip, with the side chain of Leu25 being inserted between the αTM1a$_{1-14}$Zip chains. The N-terminal residues Ser2-Ile16 of Lmod2s1 are disordered (flexible) and interact with αTM1a$_{1-14}$Zip only transiently. Inside the disordered region, residues Lys12-Glu14 form a transient α-helical turn.

The structure of the complex is characterized by several ionic and hydrophobic interactions between Lmod2s1 and αTM1a$_{1-14}$Zip. Sequence-specific ionic interactions between tropomyosin and Lmod2 chains are between side chains of Asp19 (Lmod2s1) and Lys5 (αTM1a$_{1-14}$Zip), Glu36 (Lmod2s1) and Lys5 (αTM1a$_{1-14}$Zip), Glu38 (Lmod2s1) and Lys7 (αTM1a$_{1-14}$Zip), and Glu41 (Lmod2s1) and Lys12 (αTM1a$_{1-14}$Zip). In members of the Tmod family, Asp19 is a highly conserved residue, and positions corresponding to 36, 38, and 41 of Lmod2

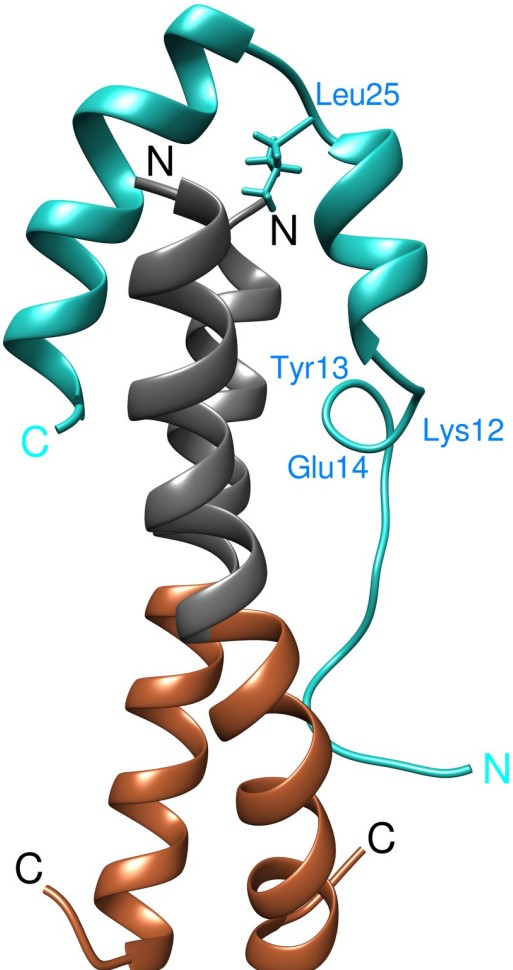

**Fig 1. Ribbon representation of the 3D structure of the Lmod2s1/αTM1a$_{1-14}$Zip complex (PDB ID 6UT2).** N- and C-termini of Lmod2s1 and αTM1a$_{1-14}$Zip are marked. The ribbon of Lmod2s1 is shown in cyan. The N-terminal Gly residue and the first 14 residues of Tpm1.1 in αTM1a$_{1-14}$Zip are colored in black, and the C-terminal GCN4 sequence is colored in brown. The side chain of Leu25 in the link motif connecting two Lmod2 helices is shown in cyan and labeled. The residues of a transiently forming loop Lys12-Glu14 in Lmod2s1 are also labeled. 3D, three-dimensional; Lmod, leiomodin; PDB, Protein Data Bank.

have a clear preference for negatively charged Glu or Asp residues (S2 Fig). Important residues contributing to the hydrophobic "cap" cluster are Leu22, Leu25, Leu30, Leu33, and Leu37 of Lmod2s1 and Met1/Ile4/Met8 from two chains of $\alpha$TM1a$_{1-14}$Zip. Residues Leu22, Leu25, and Leu33 are highly conserved in Lmod/Tmod homologs, suggesting that the geometry of a Leu side chain in these positions is critical for favorable interactions. Leu30 and Leu37 allow for a Met substitution, which is a very similar residue in hydrophobicity and size. Consistently, Leu30 was shown to be a physiologically significant residue for Lmod2, and the replacement of Leu30 in Lmod2 with a negatively charged Glu disrupted the tropomyosin-dependent ability of an Lmod2 fragment (residues 1–201) to decrease the pointed-end polymerization of actin. In addition, the replacement also disrupted Lmod2 assembly at the pointed end of the thin filaments in cardiomyocytes [23].

**The 3D structural information for the Lmod2s1/$\alpha$TM1a$_{1-14}$Zip complex was obtained using NMR spectra and TALOS+.**   To obtain the structure in Fig 1, two NMR samples of the Lmod2s1/$\alpha$TM1a$_{1-14}$Zip complex were used. In the samples, either Lmod2s1 or $\alpha$TM1a$_{1-14}$Zip were $^{15}$N/$^{13}$C-labeled, and their counterparts were unlabeled. $^{15}$N-heteronuclear single-quantum coherence (HSQC) spectra of the respective samples with sequence-specific cross-peak assignments are shown in S1 Fig. For chemical shift assignment, we recorded a series of 3D triple-resonance NMR spectra. In the spectra, we observed line broadening (due to the exchange between associated and dissociated species) and significant side-chain resonance overlap. The overlap was caused by disproportionally high occurrence of Leu, Glu, and Lys residues in the sequences. As a result of the exchange, many side-chain cross-peaks broadened beyond detection, the side-chain assignment in the complex was incomplete, and calculation of the structure based on nuclear Overhauser effect (NOE) distance restraints was not feasible (see Materials and methods, sections "NMR sample preparation" and "NMR assignment and analysis" for more details). Therefore, we utilized the dependence of $^{15}$N, $^{13}$C, and $^{1}$H chemical shift values on the local protein backbone conformation [31] and determined the backbone dihedral angles ($\varphi$,$\psi$) using TALOS+ (S1 Table). To identify flexible and well-structured parts of the complex, dynamic behavior of residues was determined on the basis of random coil index (RCI)-based estimation of the model-free order parameters (RCI-$S^2$) (S1 Table).

**Upon binding to $\alpha$TM1a$_{1-14}$Zip, the N-terminal part of Lmod2s1 remains dynamic, while its C-terminal part forms two $\alpha$-helices.**   On the basis of RCI-$S^2$ values, the N-terminal Phe4-Ile16 residues of Lmod2s1 were identified as dynamic residues whereas C-terminal residues Asp17-Asp39 were well structured. Inside the dynamic region, residues Lys12-Ile16 displayed more restricted motions. In the well-structured region (Asp17-Asp39), residues Glu18-Ser24 and Ala27-Glu38 formed two $\alpha$-helices referred to as the first and the second helices, respectively. The helices were separated by a loop formed by two residues, Leu25 and Ser26. We assigned the loop to a $\beta\beta$-loop motif, the most common two-residue motif connecting two $\alpha$-helices [32]. The $\beta$ conformation of the loop residues positioned the two helices at an angle with respect to each other (S3A Fig). Using UCSF Chimera, the angle between the axes of the two helices was measured at approximately 47°.

For $\alpha$TM1a$_{1-14}$Zip, we confirmed that residues Asp2-Leu28 form a continuous $\alpha$-helix (S3B Fig). This finding was consistent with $\alpha$TM1a$_{1-14}$Zip forming a coiled coil, since the Lmod2s1/$\alpha$TM1a$_{1-14}$Zip complex is assembled from one chain of Lmod2s1 and two chains of $\alpha$TM1a$_{1-14}$Zip [27]. Formation of the coiled coil by $\alpha$TM1a$_{1-14}$Zip was expected from the peptide design and previously demonstrated using NMR for the peptide in a free form [33] or in a complex with a tropomyosin C-terminal fragment [34].

**The structural model of the Lmod2s1/$\alpha$TM1a$_{1-14}$Zip complex was refined using MDSs with NMR restraints.**   The stoichiometry of the Lmod2s1/$\alpha$TM1a$_{1-14}$Zip complex is 1:1 [27], and therefore, it consists of a total of four helices—two from the coiled coil of $\alpha$TM1a$_{1-14}$Zip

and two from a single chain of Lmod2s1. Schematically, there are two possible ways to combine the helices into a four-helix bundle. One topology would have two helices of Lmod2s1 form one side of the bundle while two helices of $\alpha$TM1a$_{1\text{-}14}$Zip form another (S4A Fig). Another topology is a crisscross topology, in which two Lmod2s1 helices (and $\alpha$TM1a$_{1\text{-}14}$Zip helices) are in the opposite corners of the bundle (S4B Fig).

The coiled coil of $\alpha$TM1a$_{1\text{-}14}$Zip has an axis of 2-fold rotational symmetry [33]. Therefore, if we assume that the topology in S4A Fig is viable, we could expect that either of the two sides of $\alpha$TM1a$_{1\text{-}14}$Zip can interact with an Lmod2s1 molecule. This could potentially lead to the formation of a 2:1 Lmod2s1:$\alpha$TM1a$_{1\text{-}14}$Zip stoichiometric complex, in contradiction to the 1:1 stoichiometry that we previously established [27]. We performed MDSs by starting with the "side-by-side" packing of the complex where the Ala27-Glu38 helix was either in approximately parallel or antiparallel orientation with respect to the $\alpha$TM1a$_{1\text{-}14}$Zip helices. Consistent with our reasoning, neither arrangement produced a stable structure, and they quickly lost structural integrity (S1–S4 Videos). In contrast to the side-by-side topology, upon the evaluation of the stability of the four-helix crisscross topology by MDS, a stable complex between Lmod2s1 and $\alpha$TM1a$_{1\text{-}14}$Zip was produced (S5 Video).

For a quantitative comparison of the side-by-side and crisscross topologies, time evolution of two metrics during the MDS runs were calculated (shown in S5 Fig for residues Asp17-Asp39 of Lmod2s1). The two metrics were the rolling root-mean-square deviation (RMSD) between the current structure and the structure observed 1 ns before the current, and the RMSD of backbone dihedral ($\varphi,\psi$) angles calculated between the current structure and the TALOS+-predicted values. Since $\alpha$TM1a$_{1\text{-}14}$Zip forms a stable coiled coil and remained little changed in most simulations, we focused on Lmod2s1 to test the behavior of the metrics. S5A and S5B Fig demonstrate that Lmod2s1 in structures representing the side-by-side topology had larger intrinsic backbone motions (manifested as larger mean rolling RMSD and larger fluctuation [standard deviation] of rolling RMSD for each trajectory). Importantly, the RMSDs of backbone dihedral ($\varphi,\psi$) angles in Lmod2s1 (S5C Fig) for side-by-side structures were approximately 2.5-fold larger than those for crisscross helical packing, implying that the side-by-side models poorly fit the experimental NMR data.

The crisscross Lmod2s1/$\alpha$TM1a$_{1\text{-}14}$Zip complex maintained its structural integrity for the entire 400 ns used in production runs. The finalized model was validated by RDC (see Materials and methods, section "Model validation by RDC" for more details).

## The 3D structure is in agreement with spectral changes in $^{15}$N-Lmod2s1 caused by chemical/sequence alterations in the model tropomyosin peptide

**The $^{15}$N-Lmod2s1 resonance peak shifts caused by alterations in the N- and C-terminal parts of the model tropomyosin peptide are consistent with the Lmod2s1/$\alpha$TM1a$_{1\text{-}14}$Zip topology.** To enable labeling of the tropomyosin peptide for the NMR experiments, native acetylation of the tropomyosin N-terminus was mimicked by adding Gly at the N-terminus of $\alpha$TM1a$_{1\text{-}14}$Zip (see Materials and methods, section "Peptide design and production" for more details). We compared $^{15}$N-HSQC spectra of $^{15}$N-Lmod2s1 in complex with either $\alpha$TM1a$_{1\text{-}14}$Zip or Ac-$\alpha$TM1a$_{1\text{-}14}$Zip (S6 Fig). Ac-$\alpha$TM1a$_{1\text{-}14}$Zip differed from $\alpha$TM1a$_{1\text{-}14}$Zip in that its N-terminus did not contain the mimetic Gly residue, and it was N-acetylated. Since the substitution of an acetyl group with a Gly residue has very little effect on structure and binding properties of the tropomyosin peptide [35], a comparison of $^{15}$N-Lmod2s1 spectra in respective complexes identified residues of Lmod2s1 located close to the tropomyosin N-terminus.

The amide cross-peaks that shifted considerably and did not partially overlap with peaks in the original spectrum upon the replacement of $\alpha$TM1a$_{1\text{-}14}$Zip with Ac-$\alpha$TM1a$_{1\text{-}14}$Zip are

shown on the Lmod2s1 sequence and the 3D structure in Fig 2A and 2C. These residues tend to be closer to the middle of the peptide sequence, and they are mainly located in the first α-helix and the first half of the second α-helix (Fig 2A). In accordance with the 3D structure of the complex, mapping of the most affected residues onto the 3D structure demonstrated that larger spectral changes were observed for residues located in the proximity of the αTM1a$_{1-14}$Zip N-terminus (Fig 2C).

To map onto Lmod2s1 the spectral effects of changes in the C-terminus of the tropomyosin peptide, we used a previously reported comparison between the $^{15}$N-HSQC spectra of $^{15}$N-Lmod2s1 in complexes with either αTM1a$_{1-14}$Zip or αTM1a$_{1-28}$Zip [27]. The mapping of the considerably shifted cross-peaks onto the Lmod2s1 sequence and the 3D structure of the complex is shown in Fig 2B and 2D. Again, in agreement with our 3D structure, the most affected residues are located at the N-terminus of Lmod2s1—which was identified as mobile and potentially forming transient interactions with αTM1a$_{1-14}$Zip—and in the C-terminal half of the second Lmod2s1 α-helix. The most affected residues in the second helix are either Leu residues contributing to the hydrophobic cluster (Leu33 and Leu37) or their neighbors (Glu36, Asp39). This long-range effect of tropomyosin sequence extension on the second helix of Lmod2s1 can be explained by an impaired hydrophobic core in αTM1a$_{1-28}$Zip, which is less stable than αTM1a$_{1-14}$Zip [36, 37].

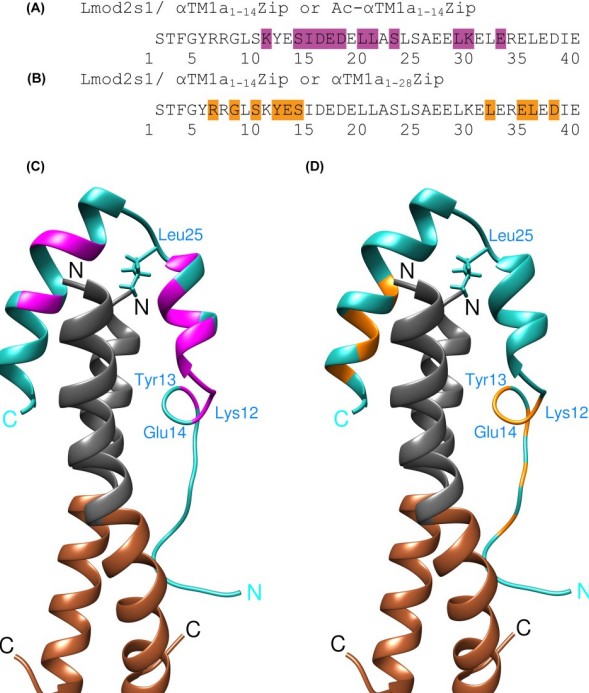

**Fig 2. Mapping of $^{15}$N-Lmod2s1 resonance peak shifts caused by chemical/sequence alterations in the model Tpm peptide onto the Lmod2s1 sequence and the 3D structure of the Lmod2s1/αTM1a$_{1-14}$Zip complex.** (A) Shown on a magenta background are $^{15}$N-Lmod2s1 residues with the most affected $^{15}$N-HSQC cross-peaks upon the replacement of the N-terminal Gly residue in αTM1a$_{1-14}$Zip with an acetyl group. (B) Shown on an orange background are $^{15}$N-Lmod2s1 residues with the most affected $^{15}$N-HSQC cross-peaks upon the replacement of αTM1a$_{1-14}$Zip (containing 14 N-terminal residues of Tpm) with αTM1a$_{1-28}$Zip (containing 28 N-terminal residues of Tpm). (C) Mapping of the most affected regions of $^{15}$N-Lmod2s1 (shown in magenta) upon replacement of the N-terminal Gly residue in αTM1a$_{1-14}$Zip with an acetyl group onto the 3D model of the complex. (D) Mapping of the most affected residues of $^{15}$N-Lmod2s1 (shown in orange) upon replacement of αTM1a$_{1-14}$Zip with αTM1a$_{1-28}$Zip onto the 3D model of the complex. 3D, three-dimensional; HSQC, heteronuclear single-quantum coherence; Lmod, leiomodin; Tpm, tropomyosin.

**The structure of the Lmod2s1/αTM1a$_{1-14}$Zip complex reflects a complete binding interface between Lmod2 and tropomyosin.** To test if the inclusion of 28 tropomyosin amino acids into the model peptide [27] can induce the formation and binding of an extra helical structure within the N-terminal dynamic residues of Lmod2s1, we titrated $^{15}$N-Lmod2s1 with αTM1a$_{1-28}$Zip. NMR peak attenuation due to the exchange between bound and unbound $^{15}$N-Lmod2s1 was followed as a function of sub-stoichiometric αTM1a$_{1-28}$Zip concentration (S7 Fig). In this type of NMR titration experiments, the peak attenuation increases with the increase in chemical shift difference between bound and unbound states [38]. Larger peak attenuations would occur for Lmod2s1 residues if they participated in formation of a helix and/or multiple nontransient direct contacts. Fig 3 illustrates changes in $^{15}$N-Lmod2s1 spectra upon binding with either αTM1a$_{1-14}$Zip or αTM1a$_{1-28}$Zip when they were mapped onto the Lmod2s1 amino acid sequence.

Upon the titration of Lmod2s1 with αTM1a$_{1-28}$Zip, major peak attenuations were observed for the region Glu18-Ile40, whereas peak attenuations in the N-terminal part of Lmod2s1 were on average considerably smaller. Similarly, upon binding to the shorter αTM1a$_{1-14}$Zip, the region Glu18-Ile40 experiences the largest chemical shift changes upon binding, whereas the chemical shifts of the Lmod2s1 N-terminus are less affected (S8 Fig). Residues Glu18-Ile40 correspond to the well-structured part of Lmod2s1 in complex with the shorter αTM1a$_{1-14}$Zip. These similarities in spectral changes upon binding to either shorter αTM1a$_{1-14}$Zip or longer αTM1a$_{1-28}$Zip demonstrate that no additional secondary structure is formed in Lmod2s1 if more tropomyosin amino acid residues are included in the model peptides.

Weaker spectral effects in the N-terminal part of the Lmod2s1 peptide were also found to display some similarities for these two complexes. Specifically, within the N-terminal region, cross-peaks corresponding to residues Ser11-Lys12 were more affected by αTM1a$_{1-28}$Zip titration than cross-peaks corresponding to their neighbors. Consistently, if compared with other residues from the Ser2-Ile16 region, Lys12-Glu14 displayed larger chemical shifts upon binding to αTM1a$_{1-14}$Zip. Interestingly, the region Lys12-Glu14 was found to form an α-helical turn capable of transient interactions with αTM1a$_{1-14}$Zip in MDSs.

Taking these results together, we can conclude that αTM1a$_{1-14}$Zip provides the major part of the binding interface. Indeed, upon increasing the number of tropomyosin residues from 14 to 21, the binding affinity increased 2-fold at 30°C [27], which roughly corresponds to a free energy change of −0.4 kcal/mol. The relatively small change in Lmod2s1 binding energetics likely arose from a small number of transient interactions between the flexible Ser2-Ile16 region of Lmod2s1 and the longer tropomyosin peptide [27].

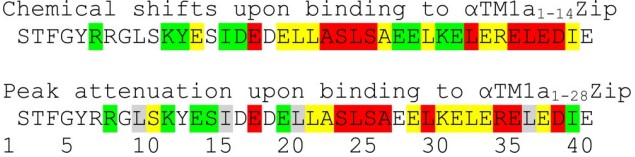

**Fig 3. Schematic comparison of the effects of αTM1a$_{1-14}$Zip and αTM1a$_{1-28}$Zip on the $^{15}$N-HSQC spectrum of $^{15}$N-labeled Lmod2s1 mapped onto the Lmod2s1 amino acid sequence.** Compound chemical shift changes (Δδ) upon Lmod2s1/αTM1a$_{1-14}$Zip complex formation were calculated as $\Delta\delta = [(\Delta\delta_H)^2 + (\Delta\delta_N/6.5)^2]^{1/2}$ [79], where $\delta_H$ and $\delta_N$ are chemical shifts of $^1$H/$^{15}$N nuclei of backbone amides. Peak attenuation upon the titration of Lmod2s1 with αTM1a$_{1-28}$Zip (data for 7:1 Lmod2s1: αTM1a$_{1-28}$Zip molar ratio were used) was calculated as a relative intensity of a $^1$H-$^{15}$N cross-peak in comparison with the intensity of the corresponding peak in the absence of αTM1a$_{1-28}$Zip. Each set of data was divided into four groups by calculating quartile values and displayed as"no color," green, yellow, and red in the order of increasing spectral effects. Since cross-peaks from pairs of Lmod2s1 residues L10/L37 and I16/L21 overlap (see Supplementary Materials in [27]), the effect of αTM1a$_{1-28}$Zip titration on their individual resonance peaks could not be determined. Hence, the residues L10, I16, L21, and L37 are labeled in the bottom sequence as gray. Lmod, leiomodin; HSQC, heteronuclear single-quantum coherence.

## Effects of reducing Lmod2 affinity for tropomyosin on Tmod1 assembly and the thin filament length in cardiomyocytes

**L25G mutation decreases affinity of Lmod2s1 for the tropomyosin fragment.** Leu25 is the first residue in a ββ-motif link between two α-helices, where it is a highly preferred residue [32]. However, one out of 10 best TALOS+-predicted conformations of Leu25 is a left-handed helical conformation ($\alpha_L$), which is highly unfavorable for Leu [39] and was typically transitioned in MDS runs into a β conformation within 1 ns. The conversion of Leu25 from a metastable $\alpha_L$ to a stable β conformation also leads to the rotation of the Leu side chain from an outward (pointing away from the $\alpha TM1a_{1\text{-}14}Zip$ N-terminus) to an inward position, where it becomes a part of the hydrophobic cluster stabilizing the $Lmod2s1/\alpha TM1a_{1\text{-}14}Zip$ complex (Fig 1). An $\alpha_L$ conformation of the first residue in two-residue links between two α-helices is typical of Gly [39]. Therefore, if, despite creating a local conformational strain, Leu25 was in an $\alpha_L$ conformation in the complex, substituting Leu25 for Gly would not affect hydrophobic interactions and could potentially increase binding due to the replacement of Leu with a considerably more favorable residue. On the other hand, for Leu25 in the β conformation, this mutation would lead to a considerable decrease in binding. To test the effect of this mutation on binding to the tropomyosin peptide, we expressed and purified a mutated Lmod2s1[L25G] peptide.

When Lmod2s1 binds to a tropomyosin model peptide, Lmod2s1 becomes more α-helical and the complex is more thermally stable than the tropomyosin coiled-coil peptide alone [26]. To establish the differences between Lmod2s1 and Lmod2s1[L25G] affinities for $\alpha TM1a_{1\text{-}14}Zip$, we tracked thermally induced unfolding by circular dichroism (CD) at 222 nm (S9 Fig). Lmod2s1 binding has a pronounced effect on the complex thermal unfolding, which previously allowed us to estimate its dissociation constant ($K_d$) as approximately 1 μM [26]. The L25G mutation decreases binding affinity of Lmod2s1 to the extent that makes binding of the mutant virtually undetectable using this approach, which is consistent with the Leu25 conformation shown in Fig 1.

**Decrease of Lmod2 binding to tropomyosin at the pointed end perturbs thin filament elongation and increases Tmod1 assembly.** The presence of TpmBS1 in Lmod2 is crucial for its proper localization to the pointed ends in cardiomyocytes [40]. To study how the L25G mutation affects the function of Lmod2 in cells, rat neonatal cardiomyocytes were transfected with green fluorescent protein (GFP), GFP-Lmod2, or GFP-Lmod2[L25G], and thin filament lengths were measured (Fig 4). GFP-Lmod2 expression increased thin filament lengths compared with the expression of GFP alone (GFP-Lmod2, 0.81 ± 0.01 μm; GFP, 0.77 ± 0.01 μm; Fig 4B), which was consistent with previous reports [5, 23, 41]. However, the introduction of

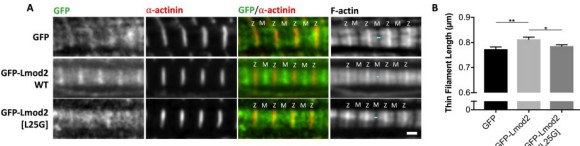

**Fig 4. GFP-Lmod2[L25G] is unable to elongate thin filaments compared with GFP-Lmod2 WT.** (A) Representative images of rat neonatal cardiomyocytes expressing GFP, GFP-Lmod2 WT, or GFP-Lmod2[L25G]. GFP-Lmod2[L25G] was unable to localize to the pointed ends, near the M-line ("M") of sarcomeres, unlike GFP-Lmod2 WT. Cells were stained with phalloidin to mark F-actin and α-actinin to indicate the Z-disc ("Z"). Turquoise lines show a gap in F-actin staining across the M-line. Scale bar = 1 μm. (B) Thin filament lengths from rat neonatal cardiomyocytes transfected with GFP, GFP-Lmod2 WT, or GFP-Lmod2[L25G] (S1 Data). Statistical data are shown as mean ± SEM, $n$ = 49–78 total measurements from 10–20 cells per culture, three cultures; $^*p$ = 0.0177, $^{**}p$ = 0.0013, one-way ANOVA. F-actin, filamentous actin; GFP, green fluorescent protein; Lmod, leiomodin; WT, wild type.

the L25G mutation perturbed the ability of Lmod2 to elongate thin filaments (GFP-Lmod2 [L25G], 0.79 ± 0.01 μm, $p$ = 0.0177, compared with GFP-Lmod2, Fig 4B). The difference between thin filament lengths in the presence of GFP (control) and GFP-Lmod2[L25G] was not statistically significant ($p$ = 0.4690). Additionally, GFP-Lmod2[L25G] was not detected (i.e., it did not assemble properly) at the pointed ends of the thin filaments, unlike GFP-Lmod2 (Fig 4A).

We have previously shown that expression of Lmod2 leads to displacement of Tmod1 from the pointed ends in cardiomyocytes [21]. Here, we examined the effect of GFP-Lmod2[L25G] on thin filament pointed-end localization of endogenous Tmod1 (Fig 5). About 90% ± 4% of control GFP-expressing cells showed consistent Tmod1 assembly to the pointed ends of thin filaments, as expected (Fig 5B). GFP-Lmod2 expression decreased the percentage of cells displaying consistent Tmod1 assembly (GFP-Lmod2, 41% ± 10%, $p$ = 0.0011). The assembly of Tmod1 in cells expressing the GFP-Lmod2[L25G] mutant was noticeably higher (GFP-Lmod2 [L25G], 67% ± 2%, $p$ = 0.0427) than in cells expressing WT GFP-Lmod2. The difference between the assembly of Tmod1 in cells expressing GFP-Lmod2[L25G] and those expressing GFP only (control) was not statistically significant ($p$ = 0.079). Therefore, the Lmod2[L25G] mutant might not be able to displace Tmod1 from pointed ends, or, at least, it might be less effective than WT GFP-Lmod2. The cellular results demonstrate that the L25G mutation significantly perturbed the subcellular localization and function of Lmod2 in cardiac muscle cells, and this agrees with the structural importance of this residue for TpmBS1 and the critical role of TpmBS1 in Lmod2 function at the pointed end. Collectively, these data support the competition mechanism.

## A model of Lmod 3D assembly and function at the pointed end

**Lmod2 and Tmod1 binding interfaces with tropomyosin are structurally homologous.** The alignment of Lmod2s1 sequence directly interacting with tropomyosin with the corresponding TpmBS1 sequence in Tmod1 shows approximately 57% sequence identity and approximately 78% sequence similarity. Therefore, we expected that the structure of complexes formed between tropomyosin and TpmBS1 of either Lmod2 or Tmod1 are very similar. To test this, we substituted the Lmod2s1 sequence in the Lmod2s1/αTM1a$_{1-14}$Zip structure with the homologous sequence from Tmod1 and subjected the new complex to 400-ns MDS. The structure of the complex formed by Tmod1 TpmBS1 and tropomyosin remained essentially the same as that of the Lmod2s1/αTM1a$_{1-14}$Zip complex (S10 Fig). This modeling result provides evidence in favor of our proposition that the N-terminus of tropomyosin cannot accommodate a simultaneous binding of both Tmod1 and Lmod2.

**The structure of the Lmod2s1/αTM1a$_{1-14}$Zip complex bears a resemblance to the head-to-tail tropomyosin overlap region.** We compared the 3D structures of the Lmod2s1/

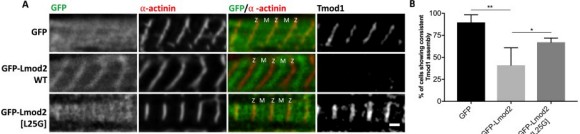

**Fig 5. GFP-Lmod2[L25G] does not significantly displace Tmod1 from thin filament pointed ends.** (A) Representative images showing Tmod1 assembly in rat neonatal cardiomyocytes expressing either GFP, GFP-Lmod2 WT, or GFP-Lmod2[L25G]. Cells were stained for Tmod1 and α-actinin (red) to mark Z. Scale bar = 1 μm. (B) Percentage of cells having consistent Tmod1 assembly at thin filament pointed ends (S2 Data). Mean ± SEM, $n$ = 124–158, total number of cells measured, four cultures (*$p$ = 0.0427, **$p$ = 0.0011, one-way ANOVA). GFP, green fluorescent protein; Lmod, leiomodin; M, M-line; Tmod, tropomodulin; WT, wild type; Z, Z-disc.

αTM1a$_{1-14}$Zip complex with the overlap complex formed by tropomyosin N- and C-termini [34]. Positioning of the two Lmod2s1 helices with respect to αTM1a$_{1-14}$Zip helices resembled that of the C-terminal fragment in the overlap complex (Fig 6), which means that TpmBS1 of Lmod2 can only bind the tropomyosin N-terminus if the latter is free and not a part of the overlap complex (at the pointed end). Comparison of the 3D structures also revealed sequence similarities between the tropomyosin C-terminus and the second helix of Lmod2s1. Among these similar residues, hydrophobic residues in these two complexes are responsible for maintaining important core contacts with the N-terminus of tropomyosin.

**A 3D reconstruction of Lmod2 assembly at the pointed end of the thin filament.** Using our newly obtained structure and currently available structural information on Lmod2/ Tmod1 interactions with actin [19, 23, 28, 42, 43], we created a model for Lmod2 assembly at the pointed end (Fig 7) using as a starting point an atomic model for F-actin in complex with one tropomyosin cable [44]. The interactions of ABS1 and ABS2 with the pointed end were modeled from the structure of the complex of homologous Tmod1 ABS1 with actin (Protein Data Bank [PDB] ID 4PKG, [19]) and from the structure of the complex of Lmod2 ABS2 with actin (PDB ID 5WFN, [42, 43]), respectively.

Although the existence of ABS1 in Lmod2 was questioned [25], we later demonstrated by NMR [23] and atomic force microscopy [28] that Lmod2 can interact with actin through this site. Currently, only the 3D structure of Tmod1 ABS1 in complex with actin is available [19]. Lmod2 and Tmod1 have ABS1 regions that may overlap only partially [23, 25]. However, within the segment of the Tmod1 sequence with the known 3D structure, Lmod2 and Tmod1 share a homologous amphipathic helix (residues Arg66-Lys79 in Lmod2) and several homologous residues in extended conformation flanking this helix [10]. We used these shared residues (Pro60-Leu86 of Lmod2) to model an approximate localization and structure of Lmod2 ABS1 on actin (Fig 7). Unlike in Tmod1, several residues of Lmod2 (residues Asn45-Arg51) located

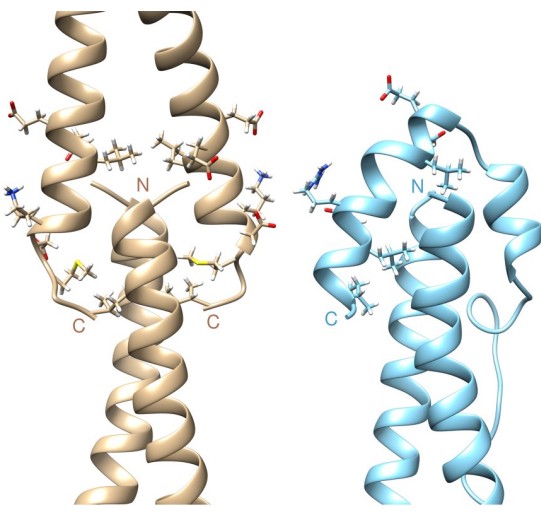

```
AEELKELERELEDIE    C-terminal helix of Lmod2s1
SEELDHALKDMTSI     C-terminus of Tpm1.1
```

**Fig 6. Comparison of the head-to-tail Tpm overlap junction (PDB ID 2G9J, left) with the Lmod2s1/αTM1a$_{1-14}$Zip complex (right).** The sequence alignment of the structurally homologous parts of the C-terminal helix of Lmod2s1 and C-terminus of Tpm1.1 is shown at the bottom of the figure. Similar residues are marked in green on the sequences and their side chains are displayed on the 3D structures. 3D, three-dimensional; Lmod, leiomodin; PDB, Protein Data Bank; Tpm, tropomyosin.

before the shared ABS1 region, are critical for Lmod2 ABS1 binding and might also interact with actin [23]. Since the full-size ABS1 of Lmod2 might include these additional residues, for the purposes of further discussion, we will henceforth call the shared homologous residues Pro60-Leu86 of Lmod2 as "ABS1h."

To complete the docking of Lmod2 to the pointed end, we connected the binding sites with polypeptide linkers (residues Pro42-Thr59 to connect TpmBS1 and ABS1h, and residues Gly87-Asn195 to connect ABS1h and ABS2). The structures of these linkers and their mode of interaction with the pointed end are not known. Accordingly, to check if the linkers were sufficiently long to support our proposed 3D structure of the assembly, we tested if the linkers form secondary structure and measured their contour lengths. For that, the linkers were created in silico using UCSF Chimera as fully extended polypeptides. After 50-ns MDS, both linkers form dynamic α-helices alternating with disordered regions (S11A Fig). Contour lengths of the Pro42-Thr59 and Gly87-Asn195 linkers with the secondary structure elements were estimated as 59.5 Å and 306 Å, respectively, providing enough length to connect the binding sites at the pointed end without steric clashes with the thin filament. Secondary structure predictions made by three popular server-side predictors—Jpred4, PsiPred, and PredictProtein—suggest locations of helical regions that partially overlap with the helical regions formed in the course of the MDSs (S11B Fig). Importantly, all of the predictors suggested that a smaller number of residues were involved in the formation of α-helices, therefore indicating that the contour lengths of the linkers might be even longer. The linkers were connected using UCSF Chimera with their respective flanking TpmBS1, ABS1h, and ABS2 in such a way as to not change the integrity of secondary structure elements formed during the simulations. The disordered regions were considered conformationally variable and were changed to adapt to the shape of the thin filament. The final assembly is shown in Fig 7.

The reconstructed Lmod2/pointed-end assembly demonstrates that our model of the Lmod/tropomyosin binding interface is consistent with data on the interactions of Lmod with actin [19, 23, 28, 42, 43]. It also shows that Lmod2 and Tmod1 share a large binding surface at the pointed end, which is formed by TpmBS1, ABS1, and ABS2.

**A model for the leaky cap created by Lmod2 binding at the pointed end of the thin filament.** The structural data provide a compelling model for the role of Lmod2 at the pointed end. In Fig 8, we propose steps of actin polymerization at the pointed end in the presence of bound Lmod2. As the first step of polymerization, an actin molecule (actin 1) attaches to the actin molecule B (Fig 8B). The linker connecting ABS1h and ABS2 is sufficiently long to create no clashes at the first step. For the second molecule of actin (actin 2) to attach to actin molecule A, the Lmod2 amphipathic helix (residues Arg66-Lys79) has to dissociate (Fig 8C); otherwise, it would interfere with the newly attached actin molecule 2. Interstrand interactions between actin 1 and actin 2 facilitate this displacement. Here, the helix acts as a "swinging gate" allowing further actin polymerization. Lmod2 residues Asn45-Arg51, critical for Lmod2 ABS1 binding to actin [23], do not obstruct the attachment of actin 2 to the pointed end, and therefore they may serve as a "hinge" inside the ABS1 region allowing the amphipathic helix to move away (and potentially unfold) from the site of intrastrand actin-actin interaction.

According to the model we propose, once two actin molecules attach to the pointed end/Lmod2 assembly, actin polymerization continues unrestricted until seven consecutive actin monomers on each side of the filament are added. Afterwards, two tropomyosin molecules bind to the seven monomers cooperatively on each side, and one of them displaces TpmBS1 of Lmod2.

The tropomyosin N-termini at the new pointed end of the filament become available for either Tmod or Lmod binding. Note that polymerization of actin at the pointed end in the presence of Lmod2 may not result in immediate removal of Lmod from the filament; it still

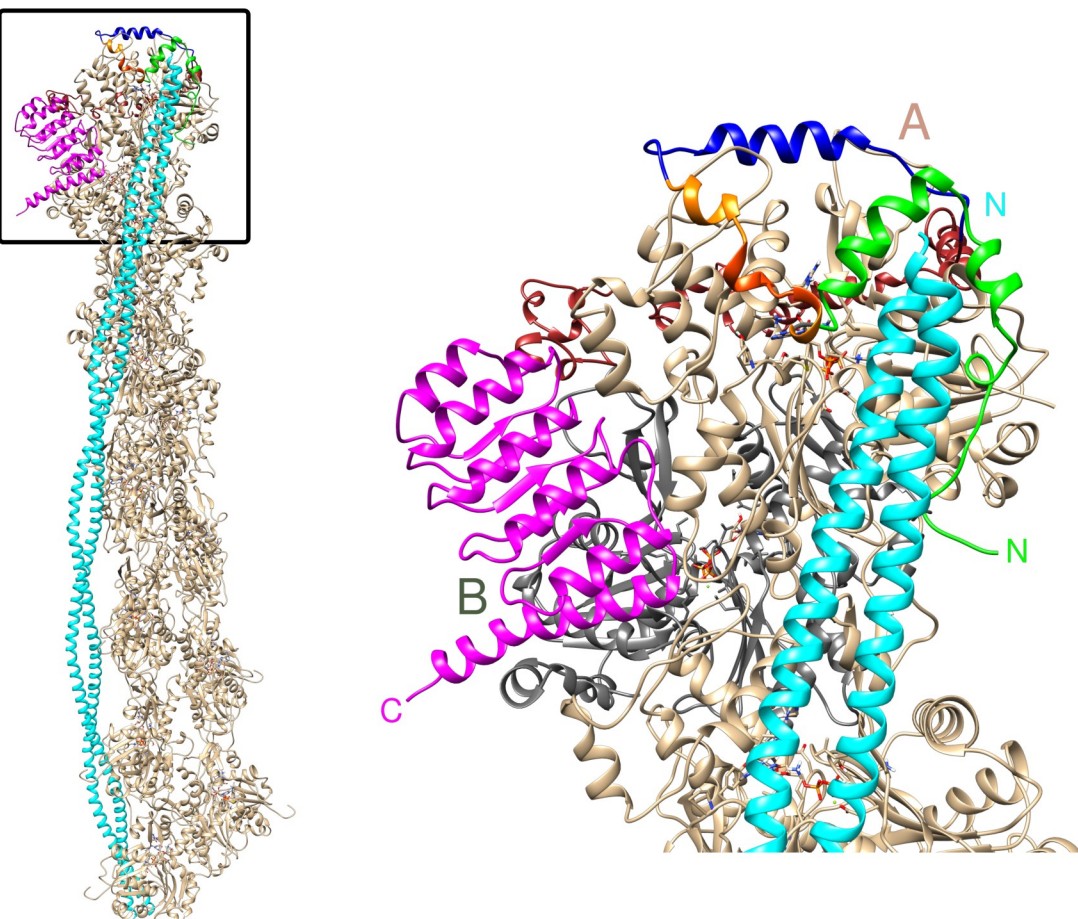

**Fig 7. A model of Lmod2 assembly at the pointed end.** The view on the left shows the Lmod2 assembly at the pointed end of the thin filament including the entire tropomyosin molecule bound to F-actin (S3 Data). The view on the right shows the magnified boxed part of the assembly. Letters A and B label two pointed-end actin molecules (colored in tan and gray, respectively, to facilitate interpretation of the figure). Tropomyosin is shown in cyan, actin (with the exception of the pointed-end actin protomer B in the right panel) in tan, and TpmBS1 of Lmod2 in green. The Lmod2 ABS1h fragment shared with Tmod1 (residues Pro60-Leu86 in Lmod2) is shown in blue (modeled from the complex of homologous Tmod1 ABS1 with actin, PDB ID 4PKG, [19]). The fragment includes the conservative amphiphatic helix (residues Arg66-Lys79 in Lmod2). Within the linker connecting TpmBS1 and ABS1h (residues Pro42-Thr59), residues Asn45-Arg51, which were found critical for Lmod2 ABS1 binding to actin [23], are shown in red-orange. The remaining linker residues are shown in orange. ABS2 (LRR) is shown in magenta (modeled from the complex of Lmod2 LRR with actin [PDB ID 5WFN, [42, 43]]). The linker connecting ABS1h and ABS2 (residues Gly87-Asn195) is in the back and shown in brown. ABS, actin-binding site; F-actin, filamentous actin; Lmod, leiomodin; LRR, leucine-rich repeat; PDB, Protein Data Bank; TpmBS1, tropomyosin-binding site.

can be bound through other actin-binding sites. This observation provides an explanation for why, unlike Tmod1, Lmod2 is found not only at the pointed end but also along the filament near the pointed end [21, 40].

## Conclusions

Members of the Tmod family are major regulators of actin filament lengths and dynamics in both muscle and nonmuscle cells [8, 9, 45]. The binding of Tmod/Lmod to the thin filament occurs via several synergistic sites, each being either actin- or tropomyosin-specific [9, 46]. To date, only partial structural information is available on the mode of binding between Tmod/Lmod and actin filaments, and only with respect to actin-specific binding [19, 42, 43].

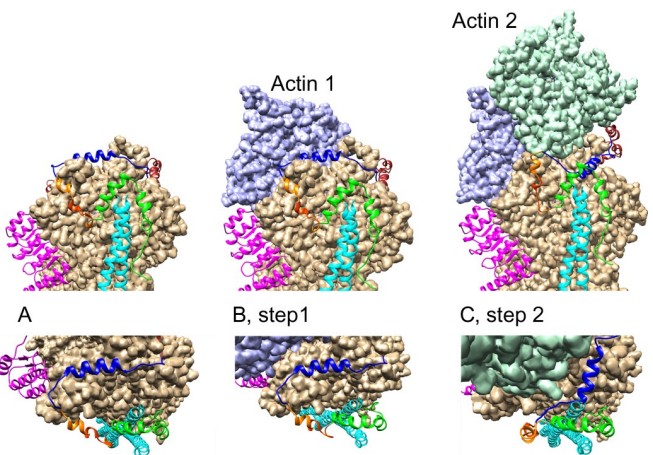

**Fig 8. A model for the leaky cap created by Lmod2 binding at the pointed end of the thin filament.** Top figures show side views, and bottom figures display views from the top. (A) The initial position of Lmod2 assembled at the pointed end and corresponding to the structure shown in Fig 7. ABS1 is in a "closed gate" conformation with the blue helix of ABS1 obstructing the attachment of an actin molecules to actin A. (B) Step 1, an actin molecule 1 (shown in purple) attaches to actin B. Actin molecule 1 does not interfere with the Lmod2 binding, and the ABS1 of Lmod2 remains in the "closed gate" conformation. (C) An actin molecule 2 (shown in turquoise) attaches to actin A. Interstrand interactions between actin molecules 1 and 2 facilitate displacement of Lmod2 ABS1 helix, and ABS1 adopts an "open gate" conformation (the blue helix moves away from the point of intrachain actin-actin interaction interface). This allows further actin polymerization until new tropomyosin N-termini are available for Lmod or Tmod attachment. ABS, actin-binding site; Lmod, leiomodin; Tmod, tropomodulin.

Understanding the mode of tropomyosin-specific binding, either via TpmBS1 or via TpmBS2, has remained elusive, until this report.

This study provides—for the first time, to our knowledge—a 3D model for the TpmBS1/Tpm1.1 molecular interface. This mode of TpmBS1 binding is only possible at the pointed end of actin filaments where the N-terminus of tropomyosin does not interact with the C-terminus of an adjacent tropomyosin molecule. TpmBS1 is the only tropomyosin-binding site that is universally present in all the members of the Tmod family. A high degree of sequence homology in TpmBS1 between all Lmod/Tmod isoforms suggests that this site plays an important role in Tmod and Lmod tropomyosin-specific recognition at actin filament pointed ends.

Importantly, the sequence comparison between different Tmod and Lmod proteins is in good agreement with the 3D model of the complex. S2 Fig shows TpmBS1 sequences of Lmod2 from different species and TpmBS1 sequences of Lmod/Tmod isoforms. Most of the residues that form the TpmBS1/Tpm1.1 binding interface are either conserved (highlighted in gray) or are conservative substitutions of similar volume and charge/hydrophobicity (highlighted in green). Occasional nonconservative substitutions at the binding interface (highlighted in yellow) may be useful for isoform-specific tuning of binding affinity [26, 36, 46]. The majority of sequence variability is observed for residues that are not in contact with Tpm1.1 (substitutions are highlighted in cyan). The formation of the N-terminal hydrophobic "cap" cluster is consistent with the report that tropomyosin peptides with positively charged unacetylated N-terminus poorly bind Tmod [35].

The debate about the regulatory role of Lmod2 at the pointed end has been centered on the interaction of Lmod2 with tropomyosin via TpmBS1 and with actin via ABS1 [9, 10, 46]. Our structural findings imply that in striated muscles, TpmBS1 of either Tmod or Lmod must bind to the same tropomyosin protomer at the pointed end and that this binding is mutually exclusive. A reconstruction of the Lmod2 assembly at the pointed end demonstrates that there is a

considerable overlap between Lmod2 and Tmod1 binding surfaces, meaning that Tmod and Lmod compete for binding and might be involved in a process of constant dynamic exchange. It was shown that there is a continuous incorporation of actin monomers at the pointed end in cardiac myocytes [47]. If thin filaments disassemble at the barbed end, their proper length could be maintained via slow and regulated actin polymerization at the pointed end when Lmod2 substitutes Tmod1. Our data suggest that Lmod2 binding at the pointed end may play an important role in actin exchange in the process of thin filament maintenance.

Similar to Tmod1, Lmod2 should prevent depolymerization from the pointed end. However, unlike in Tmod1, the ABS1 site of Lmod2 occluding the intrastrand actin-actin binding site has the flexibility to move laterally (Fig 8). This gives Lmod the ability to allow further actin polymerization turning it into a leaky cap. Interaction of the amphipathic helix in Tmod1 ABS1 with actin is stronger than that for Lmod2 [23, 25]. Moreover, ABS1 binding in Tmod1 is additionally reinforced by the second tropomyosin-binding site (TpmBS2). In this respect, Tmod's TpmBS2 acts like a latch, which fixes the helix in its position over actin molecule A, not allowing the helix to move, as opposed to the "swinging gate" helix in Lmod2.

## Materials and methods

### Ethics statement

Experiments utilizing rats were performed under the approval by The Institutional Animal Care and Use Committee at the University of Arizona (protocol # 08–017), which conforms to all applicable federal and institutional policies, procedures, and regulations, including the PHS Policy on Humane Care and Use of Laboratory Animals, USDA regulations (9 CFR Parts 1, 2, 3), the Federal Animal Welfare Act (7 USC 2131 et. Seq.), the Guide for the Care and Use of Laboratory Animals, and all relevant institutional regulations and policies regarding animal care and use at the University of Arizona.

### Peptide design and production

The choice of Lmod2s1 and αTM1a$_{1-14}$Zip amino acid sequences was described in [27]. Briefly, αTM1a$_{1-14}$Zip was designed to contain 14 N-terminal residues of Tpm1.1, followed by a GCN4 leucine zipper sequence [48] to promote formation of a coiled coil in the tropomyosin fragment. Fusing GCN4 leucine zipper sequences with tropomyosin fragments is a common approach in structural studies of tropomyosins and their complexes [34, 49]. To enable the peptide expression and uniform isotope labeling in *Escherichia coli*, the αTM1a$_{1-14}$Zip peptide also included an N-terminal Gly residue. The substitution of an acetyl group with a Gly residue was demonstrated to create a good mimetic for both structural and functional studies of tropomyosin N-terminal fragments and their complexes [34, 35, 50].

Synthetic peptides were synthesized in Tufts University Core Facility (Boston, MA). $^{15}$N- and $^{15}$N/$^{13}$C-labeled human Lmod2s1 peptides were expressed as described previously [27]. $^{15}$N- and $^{15}$N/$^{13}$C-labeled Tpm1.1 peptides were expressed following essentially the same protocol as the expression of labeled Lmod2s1. The expression minimal medium contained 50 mM Na$_2$HPO$_4$, 50 mM KH$_2$PO$_4$ (adjusted to pH 7.0 with HCl), 30 mM NaCl, 3.7 g/L $^{15}$N-ammonium sulfate, 4 g/L nonlabeled or $^{13}$C-glucose, 2 mM MgSO$_4$, 1X BME vitamins (Sigma-Aldrich, St. Louis, MO), and trace elements (20 μM FeCl$_3$, 4 μM CaCl$_2$, 2 μM ZnSO$_4$, 0.4 μM CoCl$_2$, 0.4 μM CuSO$_4$, 0.4 μM NiCl$_2$, and 0.4 μM H$_3$BO$_3$), which are known to promote the growth of BL21(DE3) *E. coli* cells [51]. The first rounds of purification of recombinant human Lmod2s1 and human Tpm1.1 peptides were described in [27]. When necessary, the peptides were additionally purified by reversed-phase HPLC on an Xbridge C18 5 μm 4.6 × 250-mm column (Waters, Milford, MA) to eliminate minor impurities. In the second round of

purification, Lmod2s1 was purified in a linear 1%/minute ammonium carbonate/acetonitrile gradient, whereas Tpm1.1 peptides were purified in a linear 1%/minute 100 mM sodium phosphate (pH 11.2)/acetonitrile gradient. The peptides were acidified by addition of 10 μL 6 M HCl per each mL of collected peaks immediately after purification and desalted on the same HPLC column or on a Sep-Pak C18 cartridge (Waters) with 0.1% trifluoroacetic acid (TFA). The desalted peptides were eluted with a 1/1 (v/v) 0.1% TFA/acetonitrile mixture and lyophilized. Peptide concentrations were determined by measuring their difference spectra at 294 nm in 6 M guanidine-HCl between pH 7.0 and 12.5 using the extinction coefficient of 2,357 cm$^{-1}$ per 1 M tyrosine [27].

The L25G mutation was introduced into Lmod2s1 by amplifying a DNA vector expressing WT Lmod2s1 [27] via PCR using partially overlapping complementary primers incorporating the mutation. The primer sequences were forward primer: 5′-CGAGC<u>GGC</u>AGCGCTGAA GAACTGAAGGAACTGGAACGC-3′, reverse primer: 5′-AGCGCT<u>GCC</u>GCTCGCCAGC AGTTCATCTTCATCAATAGA-3′, where underlined nucleotides encode for Gly instead of Leu25. The PCR reaction was performed with *PfuTurbo* DNA polymerase (Agilent Technologies, Santa Clara, CA), and the PCR product was used to transform DH5α *E. coli* cells (Life Technologies, Carlsbad, CA). Presence of the L25G mutation and absence of nonspecific mutations were confirmed by Sanger sequencing (Genewiz, South Plainfield, NJ). Expression of the mutant protein was performed in BL21(DE3) *E. coli* cells (Life Technologies, Carlsbad, CA) grown overnight in ZYP medium [51]. Purification was performed as described for WT Lmod2s1 [27].

## Plasmid construction and site-directed mutagenesis for Lmod2 and Lmod2 [L25G] expression in cardiomyocytes

Full-length mouse Lmod2 (mLmod2) was cloned from cDNA generated from mouse hearts. The coding sequence was inserted into the pEGFP-C2 vector (Clontech, Mountain View, CA) using *Xho*I and *Hin*dIII restriction sites. The L25G mutation was introduced into pEGFP-C2-mLmod2 by PCR using Phusion High-Fidelity DNA Polymerase (New England Biolabs, Ipswich, MA) and two partially overlapping complementary oligonucleotides with the following sequences: 5′-TGGCCTCC<u>GG</u>CTCACCTGAAGAGCTGAAGGAGCTTGAGAG-3′ and 5′-CAGGTGAG <u>CC</u>GGAGGCCAGAAGTTCATCCTCATCAATGGATTCA-3′. Substituted nucleotides are underlined. Generated PCR product with the mutation was treated with *DpnI* (New England Biolabs) to digest the remaining WT template and used to transform DH5α *E. coli* cells (Life Technologies, Carlsbad, CA). Kanamycin resistance clones were analyzed for presence of the mutation. Plasmids were purified using ZR Plasmid Miniprep Classic kit (Zymo Research, Irvine, CA) according to manufacturer's recommendations. DNA Sanger sequencing to confirm successful mutagenesis was performed by Eton Bioscience (San Diego, CA). Synthesis of the oligonucleotide primers was performed by Sigma-Aldrich (St. Louis, MO).

## NMR sample preparation

NMR samples were prepared by reconstituting and mixing lyophilized Lmod2s1 and Tpm1.1 peptides in 50 mM sodium phosphate buffer (pH 6.5), 10% $D_2O$, 0.2 mM EDTA, 0.1% sodium azide, and 2X Pierce EDTA-free protease inhibitor cocktail. The sample pH was adjusted by the addition of small volumes of 1 M HCl or NaOH. The pH was measured with the Mettler Toledo (Columbus, OH) U402-M3-S7/200 microelectrode. In each NMR sample, one of the two peptides in the Lmod2s1/Tpm1.1 peptide complex was labeled whereas the other one was unlabeled.

The concentration of the labeled peptide was approximately 0.5 mM. Initially, 1.5-fold stoichiometric excess of the unlabeled peptide was used. However, we found that many crosspeaks in the $^{15}$N-HSQC spectra were broad (S12A and S12B Fig). Under these stoichiometric

ratio conditions we estimated the apparent rotational correlation time $\tau_c$ of the $^{15}$N/$^{13}$C-$\alpha$TM1a$_{1-14}$Zip/Lmod2s1 complex to be roughly 17 ns [52], which exceeded more than 2-fold a typical $\tau_c$ (approximately 7.2 ns) for a rigid and approximately spherical protein of this size [53, 54]. We found that further increase of the concentration of the unlabeled peptide led to narrowing of the resonance lines (S12C Fig). Given that the dissociation constant ($K_d$) of the complex is in low µM range [27], this observation indicated that chemical exchange between bound and traces (approximately 1%) of unbound labeled peptides contributed significantly to the T2 relaxation [55]. To decrease broadening of the resonance lines that was observed as a result of the dissociation/association exchange process, most 2D and 3D NMR spectra were recorded with the unlabeled peptide being in approximately 3 times stoichiometric excess unless otherwise stated. An attempt to create higher stoichiometric excess led to severe line broadening followed by protein precipitation. Therefore, our ability to reduce the rate of T2 relaxation in the sample was limited by the solubility of the peptide components.

## NMR spectra collection

The NMR spectra were recorded at 25˚C on a Varian VNMRS 600 MHz spectrometer (Agilent Technologies, Santa Clara, CA) equipped with a 5-mm triple-resonance probe or on a Varian Inova 500-MHz spectrometer equipped with a 5-mm Nalorac Z-Spec HCNP triple-resonance probe. The complete data set included 2D $^{15}$N-HSQC and $^{13}$C-HSQC and 3D HNCO, HNCA, HN(CO)CA, HNCACB, CBCA(CO)NH, HBHA(CO)NH, HCCH-TOCSY, and $^{15}$N- and $^{13}$C-edited NOESY [56]. The NMR spectra were processed using NMRPipe [57] and Felix (Felix NMR, San Diego, CA). The 2D $^{15}$N-HSQC titration spectra were processed and analyzed with MestReNova (Mestrelab Research, S.L., Santiago de Compostela, Spain). NMRViewJ (One Moon Scientific) was used for 3D NMR spectra visualization and peak assignment [58] (BMRB ID 30681).

## NMR assignment and analysis

Almost all the backbone atom resonances were assigned. However, as a result of the association/dissociation exchange process in the complex, resonance peaks for a number of residues were broadened beyond detection in $^{13}$C-correlated 3D spectra (S13 and S14 Figs), and only partial assignment of side-chain resonances was achieved. Although not being the primary reason for the incomplete assignment, significant side-chain resonance overlap caused by disproportionally high occurrence of Leu, Glu, and Lys residues in the sequences exacerbated the problem. Because of the incomplete resonance assignment, the generation of a 3D structure for the Lmod2s1/$\alpha$TM1a$_{1-14}$Zip complex on the basis of NOE contacts [59] was not feasible. Therefore, to obtain the structure, we utilized the dependence of $^{15}$N, $^{13}$C, and $^1$H chemical shift values on the local protein backbone conformation [31] to determine the backbone dihedral angles. Protein backbone torsion angle and RCI-$S^2$ values [60] were predicted on the basis of $^{15}$N, $^{13}$C', $^{13}$C$\alpha$, $^{13}$C$\beta$, $^1$H$\alpha$, and $^1$H$^N$ chemical shifts by TALOS+ [61] using NMRViewJ-generated TALOS+ input files. TALOS+ predicts dihedral angles empirically on the basis of known NMR and X-ray data from 200 proteins.

Most of the well-structured elements in the complex represented $\alpha$-helices. In the loop, connecting the first and second helices of Lmod2s1, Ser26 was in a $\beta$ conformation, but the conformation of Leu25 was not assigned unambiguously. Out of the 10 best database matches made by TALOS+ for Leu25, nine were in the $\beta$ region of the Ramachandran plot, and one remaining match had a positive $\varphi$ value corresponding to an $\alpha_L$ (left-hand helical) conformation. Motifs $\beta\beta$ and $\alpha_L\beta$ are the most common two-residue motifs connecting two $\alpha$-helices [39]. Leu and Ser show high propensities for the first and second positions in $\beta\beta$ motifs,

respectively, whereas the $\alpha_L\beta$ conformation is favored by Gly and Asp. Leu is not likely to be found as the first residue in the $\alpha_L\beta$ motif [39], and although it was not restrained in MDSs, it was initially set to a $\beta$ conformation. Calculated backbone dihedral angles ($\varphi,\psi$) of Asp2-Leu28 in $\alpha$TM1a$_{1-14}$Zip were found in the $\alpha_R$ region (approximately $-60°,-45°$) of the Ramachandran plot, and the residues were found to be well-ordered. The RCI-$S^2$ values for these residues were found to be within the range of 0.79–0.90 (S1 Table).

To validate the use of TALOS+ for our specific complex, we tested how well TALOS+ identifies torsional angles for the same $\alpha$TM1a$_{1-14}$Zip peptide but when it was in a complex with TM9a$_{251-284}$ (PDB ID 2G9J). Chemical shifts and 3D NMR structure for the complex were determined by Greenfield and colleagues [34]. On the basis of chemical shifts, TALOS+ generated a $\alpha$TM1a$_{1-14}$Zip secondary structure similar to that in the Lmod2s1/$\alpha$TM1a$_{1-14}$Zip complex, with a continuous Asp2-Leu28 $\alpha$-helix. Residues Val29-Arg32 were not classified as part of the helix, and they displayed an increased dynamic behavior (disorder) toward the C-terminus. Val29 was determined to be in a $\alpha$-helical conformation, Gly30 was determined to be in an $\alpha_L$ conformation, whereas Glu31 was mapped to the $\beta$ region. In agreement with TALOS+, backbone relaxation parameters ($^1$H-$^{15}$N heteronuclear NOE and R1 and R2 $^{15}$N relaxation rates) determined in [34] demonstrated an increased flexibility of the C-terminal residues Gly30-Arg32. However, the results of structure calculation made by Greenfield and colleagues determined both Gly30 and Glu31 to be in the $\alpha$-helix [34]. The differences in Gly30-Glu31 conformations are likely to arise from their intrinsic dynamic behavior. Overall, for the known tropomyosin peptide structure, TALOS+ produced results in a good agreement with the reported NMR data.

## RDC values

To obtain RDC values, S3 NMR spectra [62] of the sample were recorded in 5% alkyl-PEG bicelles (n-octanol/C8E5) [63]. First, we recorded S3 NMR spectra of $^{15}$N-labeled Lmod2s1 in the presence of excess unlabeled $\alpha$TM1a$_{1-14}$Zip. By comparing $^1$H-$^{15}$N cross-peak splitting in oriented and isotropic (without bicelles) environments, we determined $^1$H-$^{15}$N RDC values for the $^{15}$N-labeled Lmod2s1 in the complex. However, we were unsuccessful in obtaining RDC values for the $^{15}$N/$^{13}$C-labeled sample of $\alpha$TM1a$_{1-14}$Zip in the presence of excess unlabeled Lmod2s1, because oriented n-octanol/C8E5 bicelles would not form for this sample. We explained this n-octanol/C8E5 behavior by its interaction with unbound Lmod2s1, which, judging by its longer retention time in the reversed-phase HPLC profile, is more hydrophobic than $\alpha$TM1a$_{1-14}$Zip and can potentially interfere with the oriented phase formation. An RDC uncertainty of 1–2 Hz was estimated from comparing RDC values determined by fitting resonance peaks in NMRViewJ and NMRFAM-SPARKY [64].

## NMR titration

For NMR titration of $^{15}$N-Lmod2s1 with $\alpha$TM1a$_{1-28}$Zip, the Lmod2s1 peptide solution (0.3 mM) was prepared in 50 mM sodium phosphate buffer (pH 6.5), 10% D$_2$O, 0.2 mM EDTA, 0.1% sodium azide, and 2X Pierce EDTA-free protease inhibitor cocktail. The pH of 70 μM $\alpha$TM1a$_{1-28}$Zip solution in water was adjusted to 6.5; aliquots were taken and lyophilized to dryness. The $^{15}$N-Lmod2s1 sample was added to dry $\alpha$TM1a$_{1-28}$Zip to create molar ratios Lmod2s1: $\alpha$TM1a$_{1-28}$Zip equal to 40:1, 30:1, 20:1, 15:1, 10:1, and 7:1. After reconstitution of $\alpha$TM1a$_{1-28}$Zip, $^{15}$N-HSQC spectra were taken on the Varian Inova 500 MHz spectrometer at 25°C.

## MDSs

Polypeptide structure editing and visualization was performed in UCSF Chimera [65]. Initial backbone dihedral angles were set to values predicted by TALOS+. The complexes were

neutralized with $Na^+$ ions and placed in a cuboid box of TIP3P water molecules [66] with a minimal distance of 10 Å from the protein complex to the edge of the box. Protonation states of histidine residues were set to neutral. The MDSs were performed using AMBER18 [67] with the ff14SB force field [68] and periodic boundary conditions. The cutoff value for nonbonded interactions was 8.0 Å (default). The pressure was set to 1 bar using a Berendsen barostat with 1-ps relaxation time (default). Energy minimization was performed with the steepest descent method for 2,500 cycles followed by the conjugate gradient method for 2,500 cycles. The temperature in MDS runs was controlled by a Langevin thermostat with a 3-ps$^{-1}$ collision frequency. Hydrogens were constrained using SHAKE. Backbone dihedral angles φ and ψ of ordered residues were restrained by values predicted by TALOS+ using dihedral potential constants rk2 = rk3 = 32.0 kcal × mol$^{-1}$ × rad$^{-1}$. Residues with restrained φ and ψ were Glu18-Ser24 and Ser26-Asp39 of Lmod2s1 and Asp2-Gly30 of αTM1a$_{1-14}$Zip. Dihedral angles of Leu25 were left unrestrained.

Depending on the starting point, initial 40-ns MDS runs at 298 K produced a few different modes of the crisscross Lmod2s1/αTM1a$_{1-14}$Zip binding, which could be generally grouped into structures with either the short (Glu18-Ser24) or the long (Ala27-Glu38) Lmod2s1 α-helix being roughly parallel to the tropomyosin coiled coil. These structures also differed in the conformation of the linker residue Leu25, which adopted either a β or an $α_L$ conformation. To converge these distinct structures, we employed a simulated annealing algorithm whereby the preliminary structures were first heated from 300 K to 360 K and then cooled back in 10-degree increments. The heating was stopped at 360 K because above 360 K, the complex lost most of its secondary structure as well as ionic and nonpolar interactions. The complex was equilibrated at each temperature for 10 ns before the temperature was changed. The time step for simulated annealing was set to 1.0 fs. The equilibration time of 10 ns was chosen based on in silico tests that showed that characteristic times of $α_L$-to-β and β-to-$α_R$ transitions of Leu25 were in the range of 1–2 ns.

To characterize the convergence quantitatively, we chose two structures—one from each group of the crisscross Lmod2s1/αTM1a$_{1-14}$Zip binding modes obtained in initial simulations—as starting structures in the simulated annealing protocol. We called the starting structure with the long (Ala27-Glu38) helix parallel to the tropomyosin coiled coil "model 1," whereas the structure with the short (Glu18-Ser24) helix parallel to the tropomyosin coiled coil was called "model 2." When backbone atoms N, Cα, and C' of well-structured residues Asp17-Asp39 (Lmod2s1) and Asp2-Leu28 (αTM1a$_{1-14}$Zip) in the two "models" were matched, RMSD between the same atoms was 3.77 Å. For the same matched structures, RMSD between the backbone atoms N, Cα, and C' of Lmod2s1 was 5.18 Å, thus reflecting the large difference between positions of Lmod2s1 in the two "models" with respect to the tropomyosin coiled coil. Seven replicas of simulated annealing were performed for each of the two starting structures. For each "model," an ensemble of seven structures was produced. Two structures in each ensemble dissociated during the heating steps, lost the majority of interstrand contacts, and failed to associate back during the cooling steps; they were consequently excluded from analysis. The remaining five structures in each ensemble were averaged and the comparison between the two average structures gave the global RMSD (calculated as above) equal to 1.36 Å and the RMSD for Lmod2s1 equal to 2.14 Å. A pairwise global RMSD value calculated for a cluster including all 10 structures from both ensembles was 2.34 ± 0.09 Å (mean ± SEM), and the respective pairwise RMSD for Lmod2s1 was 3.63 ± 0.14 Å (S2 Table). Visual inspection of all the 10 annealed structures confirmed that their topology was similar in that in all of them the long Lmod2s1 helix (Ala27-Glu38) was approximately parallel to the tropomyosin coiled coil, whereas the linker residue Leu25 adopted a β conformation and was inserted between the strands of αTM1a$_{1-14}$Zip.

One of the annealed structures was used as a starting point for twenty 400-ns MDS production runs. The temperature was set to 298 K, and the time step was set to 1 fs. Ten equilibrated complexes were subjected to energy minimization to eliminate bond length and angle distortions caused by thermal motions (PDB ID 6UT2). Global pairwise RMSD calculated for backbone atoms N, Cα, and C' of well-structured residues Asp17-Asp39 (Lmod2s1) and Asp2-Leu28 (αTM1a$_{1-14}$Zip) in the 10 structures was 1.76 ± 0.07 Å (mean ± SEM). The pairwise RMSD statistical data are attached to supporting information (S3 Table).

Discrimination between the side-by-side and crisscross packing of the helices in the complex was performed by comparing the evolution of two metrics along MDS trajectories. In the simulations, the run time was 80 ns, the temperature was set to 300 K, volume to constant, and the Langevin thermostat collision frequency was 1 ps$^{-1}$. Other MDS conditions were as above. The first metric was "rolling RMSD," which was calculated between two structures of the complex, one current and another one observed 1 ns before the current. To calculate the rolling RMSD, the two complexes were first matched over the backbone atoms N, Cα, and C' of residues Asp17-Asp39 (Lmod2s1) and Asp2-Leu28 (αTM1a$_{1-14}$Zip). After matching, the rolling RMSD value was calculated between the coordinates of the backbone atoms N, Cα, and C' of Asp17-Asp39 in Lmod2s1. The second metric was the RMSD for backbone dihedral (φ,ψ) angles between the current structure and the TALOS+ prediction (calculated for residues Asp17-Asp39 in Lmod2s1).

Simulation parameters for two linkers connecting TpmBS1 to ABS1h and ABS1h to ABS2 were the same as those for Lmod2s1/αTM1a$_{1-14}$Zip, except for those associated with temperature and pressure control. The temperature was controlled by Berendsen thermostat with a 1-ps coupling constant, and the volume was constant. The simulations for the linkers were run for 50 ns.

## Model validation by RDC

RDCs observed in anisotropic samples report information on internuclei vector orientations, and they are often used for 3D structure validation and refinement [30, 69]. We used RDC values obtained from the S3 spectra in oriented medium for the validation of the Lmod2s1/αTM1a$_{1-14}$Zip structure (S15 Fig). Consistent with the RCI-$S^2$ values determined by TALOS+, the RDC values of Phe4-Ser11 were small and did not exceed approximately 2 Hz, which is typical of dynamic disoriented residues [30]. Some backbone stiffening of Lys12-Ile16 correlated with the increase in their RDC values which range from approximately −6 to 3 Hz. The experimental RDC values for ordered residues Glu18-Glu38, which corresponded to the two Lmod2s1 helices and the connecting loop, had a good correlation with theoretical RDC values obtained by REDCAT [70] from the MDS-generated structure in Fig 1 (S16 Fig). Specifically, the RMSD was approximately 2.3 Hz, which is only slightly higher than the experimental uncertainty of 1–2 Hz, and the quality Q-factor was approximately 19%. The realistic experimental minimum of Q-factor is approximately 10%, with the best structures giving Q-factor values in the range of 10%–15%, whereas 20%–25% roughly correspond to a 1.8-Å X-ray structure [71].

## CD

CD measurements were performed using an Aviv model 420 CD spectropolarimeter (Lakewood, NJ). Spectra were recorded from 260 nm to 195 nm in 10 mM sodium phosphate buffer (pH 7.0), and 100 mM NaCl, at 0°C. Melting curves were measured at 222 nm from 0°C to 60°C with a temperature step of 0.2°C, a 0.15-°C deadband, a 0.3-min equilibration time, and 5-s averaging time. Concentrations of Lmod2s1 (WT), Lmod2s1 (L25G), and αTM1a$_{1-14}$Zip were 10 μM.

## Lmod2 assembly at the pointed end

To build the model of Lmod2 assembly at the pointed end, an atomic model for the tropomyosin cable with F-actin was used as a starting point [44]. To create a model of the pointed end available for Lmod binding, the tropomyosin molecule providing the C-terminal "tail"in the head-to-tail tropomyosin overlap complex was removed. The part of the F-actin filament that was in contact with the removed tropomyosin protomer was also removed. In the resulting complex (named Tpm-Factin), two actin molecules at the pointed end were labeled as A and B. To dock TpmBS1, ABS1h, and ABS2 (LRR) domains of Lmod2 to Tpm-Factin, first the Lmod2s1/$\alpha$TM1a$_{1-14}$Zip complex was added to Tpm-Factin via superimposing $\alpha$TM1a$_{1-14}$Zip and the N-terminus of tropomyosin using the MatchMaker tool in UCSF Chimera. The Lmod2 LRR domain was docked by superimposing actin molecule B from Tpm-Factin and the complex between LRR from Lmod2 and G-actin (PDB ID 5WFN [42], calculated using raw X-ray data obtained by Chen and colleagues [43]). As an approximation for Lmod2 ABS1h docking, we superimposed the complex of G-actin with homologous Tmod1 ABS1 (PDB ID 4PKG, [19]) with actin A of Tpm-Factin.

Two polypeptide chains connecting TpmBS1 with ABS1h and ABS1h with ABS2 of Lmod2 were subjected to 50-ns MDS runs to identify potential formation of secondary structure elements and incorporated into the model. The result of the simulation was compared with secondary structure predictions made by three server-side secondary structure predictors, Jpred4 [72], PsiPred [73], and PredictProtein [74]. The contour lengths (L) of the linkers including $\alpha$-helices and random coil residues were estimated using the formula L = 4 Å × (number of random coil residues) + 1.5Å × (number of $\alpha$-helical residues) [75]. Redundant actin molecules (from Tpm-Factin), $\alpha$TM1a$_{1-14}$Zip, nonhomologous part of Tmod1 ABS1, and gelsolin (from 4PKG) were removed. To create the final view of the assembly, we performed energy minimization using Amber18. Complexes forming at steps 1 and 2 of the putative thin filament elongation process were docked and minimized in a similar fashion.

## Isolation, transfection, and immunofluorescence microscopy of neonatal cardiomyocytes

Rat neonatal cardiomyocytes were isolated and plated on 35-mm tissue-culture dishes containing 12-mm round glass coverslips as described in [5]. Within 12 hours of plating, cells were transfected with 1 μg of either GFP, GFP-Lmod2 WT, or GFP-Lmod2 [L25G] DNA using Lipofectamine 3000 reagent (Thermo Fisher Scientific, Waltham, MA). Forty-eight hours after transfection, cells were washed with phosphate-buffered saline (PBS) and incubated in relaxing buffer (10 mM 3-[N-morpholino] propane sulfonic acid [pH 7.4], 150 mM KCl, 5 mM MgCl$_2$, 1 mM EGTA, and 4 mM ATP) for 15 min and fixed with 2% paraformaldehyde in relaxing buffer for 15 min at room temperature. Following fixation, cells were permeabilized with 0.2% Triton X-100/PBS for 20 min at room temperature and then blocked with 2% bovine serum albumin (BSA) plus 1% normal donkey serum/PBS for 1 h at room temperature. Cells were then incubated overnight at 4˚C with rabbit polyclonal anti-Tmod1 (2 μg/mL) [76] and mouse monoclonal anti-α-actinin (1:200) (EA-53; Sigma) primary antibodies. Following incubation, cells were washed with PBS for 3 × 5 min and incubated for 1.5 h at room temperature with secondary antibodies/PBS, which included Alexa Fluor 405–conjugated goat anti-mouse IgG (1:200) (Thermo Fisher Scientific), Alexa Fluor 647–conjugated donkey anti-rabbit IgG (1:300) (Jackson ImmunoResearch Laboratories, West Grove, PA), and Texas Red-X Phalloidin (Thermo Fisher Scientific). Cells were then washed with PBS for 3 × 5 min and mounted onto slides with Aqua Poly/Mount (Polysciences, Warrington, PA). Images were captured using a Nikon Eclipse T*i* microscope with a 100 × NA 1.5 objective, and a digital

complementary metal oxide semiconductor (CMOS) camera (ORCA-flash4.0; Hamamatsu Photonics, Shizuoka Prefecture, Japan). 3D deconvolution was performed using NIS offline deconvolution software (Nikon Corporation, Tokyo, Japan), and images were processed using Photoshop CC (Adobe, San Jose, CA). Thin filament lengths and sarcomere lengths were measured using the DDecon plugin for Image J [77, 78]. Thin filament lengths were analyzed between a sarcomere length range of 1.85–2.40 μm. Sarcomere lengths were not different between the groups within this range. For determining thin filament pointed-end assembly of Tmod1, positively transfected cardiomyocytes were classified as either "consistent" well-defined striated Tmod1 staining with little cytoplasmic background or "inconsistent," partial striated Tmod1 staining with high levels of cytoplasmic background.

## Statistics

All statistical analyses were performed in Prism 7.70 (Graphpad Software, San Diego, CA). Multiple groups were compared with one-way ANOVA with Tukey's post hoc test. Differences with $p < 0.05$ were considered statistically significant.

## Supporting information

**S1 Fig.** Sequence-specific assignment of 2D $^{15}$N-HSQC spectra of (A) $^{15}$N/$^{13}$C-labeled Lmod2s1 in a complex with unlabeled αTM1a$_{1-14}$Zip, and (B) $^{15}$N/$^{13}$C-labeled αTM1a$_{1-14}$Zip in a complex with unlabeled Lmod2s1. The spectra were recorded in 50 mM sodium phosphate buffer (pH 6.5), 10% D$_2$O, 0.2 mM EDTA, 0.1% sodium azide, 2X Pierce EDTA-free protease inhibitor cocktail on a Varian 500-MHz spectrometer at 25˚C. Sequences of the Lmod2s1 and αTM1a$_{1-14}$Zip peptides used in the NMR studies are shown at the bottom. The N-terminal Gly and the GCN4 sequence in αTM1a$_{1-14}$Zip are shown in small letters. 2D, two-dimensional; HSQC, heteronuclear single-quantum coherence; Lmod, leiomodin. (TIF)

**S2 Fig. Comparison of TpmBS1 sequences for Lmod2 homologs from different species and for Lmod/Tmod human isoforms.** The sequence of human Lmod2 (from this study) is used as a reference in the comparison. Conserved residues forming the TpmBS1/tropomyosin binding interface are highlighted in gray. Conservative interfacial residue replacements of similar volume [80] and charge/hydrophobicity are highlighted in green. Nonconservative interfacial replacements are highlighted in yellow. Sequence replacements of residues that are not in direct contact with tropomyosin are highlighted in cyan. Lmod, leiomodin; Tmod, tropomodulin; TpmBS1, tropomyosin-binding site. (TIF)

**S3 Fig.** 3D conformations of the Lmod2s1 (A) and αTM1a$_{1-14}$Zip (B) peptides in the Lmod2s1/αTM1a$_{1-14}$Zip complex. (A) Backbone torsion angle values (φ,ψ) of Ile16-Ile40 amino acid residues were used to build a model of Lmod2s1. The well-structured region with RCI-$S^2 \geq 0.7$ (Asp17-Asp39) is shown in blue. (B) Backbone torsion angle values (φ,ψ) of Asp2-Gly30 amino acid residues were used to build a model of the coiled-coil αTM1a$_{1-14}$Zip peptide. α-helices (Asp2-Leu28) are shown in blue. 3D, three-dimensional; Lmod, leiomodin. (TIF)

**S4 Fig. Potential schematic topologies of a four α-helix bundle assembly in the Lmod2s1/ αTM1a$_{1-14}$Zip complex prior to MDS.** Lmod2s1 helices are shown in green, and αTM1a$_{1-14}$Zip helices are shown in black. 1YO7 (ID PDB) 4-helix bundle was used as the topology template. (A) Side-by-side topology, and (B) crisscross topology. Lmod, leiomodin; MDS,

molecular dynamics simulation; PDB, Protein Data Bank.
(TIF)

**S5 Fig. Plots of the rolling RMSD and the backbone dihedral (φ,ψ) angle RMSD for rigid residues in Lmod2s1 (Asp17-Asp39) along MDS trajectories of complexes with side-by-side and crisscross helix topologies.** The rolling RMSD was calculated between two structures of the complex, one current and another one observed 1 ns before the current, as described in Materials and methods. RMSD values were averaged between two replicas. Four versions of side-by-side helix packing were tested, with two of them representing approximately parallel orientation of the C-terminal (residues Ala27-Glu38) Lmod2s1 helix with respect to the αTM1a$_{1-14}$Zip, and the other two antiparallel. For each of the two orientations of the helix, one of two sides of Lmod2s1 can interact with the αTM1a$_{1-14}$Zip. These sides were denoted as the left-hand side (when looking at Lmod2s1 in the direction from the N-terminal [residues Glu18-Ser24] to the C-terminal helices) and the right-hand side. (A) The rolling RMSD for left-hand side-by-side helical packing in comparison with the crisscross packing. The mean ± SD RMSDs (in Å) were equal to 0.76 ± 0.14 (crisscross), 1.06 ± 0.25 (parallel left-hand), and 1.22 ± 0.31 (antiparallel left-hand). (B) The rolling RMSD for right-hand side-by-side helical packing in comparison with the crisscross packing. The mean ± SD RMSDs (in Å) were equal to 1.09 ± 0.34 (parallel right-hand) and 1.08 ± 0.30 (antiparallel right-hand). (C) The (φ,ψ) RMSD between the current structure and the TALOS+ prediction. MDS, molecular dynamics simulation; RMSD, root-mean-square deviation.
(TIF)

**S6 Fig. A comparison between $^{15}$N-HSQC spectra of $^{15}$N-Lmod2s1 in complex with αTM1a$_{1-14}$Zip (shown in blue) or Ac-αTM1a$_{1-14}$Zip (shown in red).** Spectra of approximately 0.3 mM $^{15}$N-Lmod2s1 in the presence of roughly 2× excess of the corresponding unlabeled tropomyosin coiled-coiled peptide were recorded on a Varian VNMRS 600 MHz spectrometer at 25°C. The general pattern of resonance peak distribution upon the substitution of αTM1a$_{1-14}$Zip with Ac-αTM1a$_{1-14}$Zip was preserved, confirming that both complexes have a similar structure. HSQC, heteronuclear single-quantum coherence.
(TIF)

**S7 Fig. Titration of $^{15}$N-Lmod2s1 with αTM1a$_{1-28}$Zip. Green, tan, cerulean, and dark blue bars show relative cross-peak intensity I/I$_0$ of the corresponding residue in the presence of 1:20, 1:15, 1:10, and 1:7 αTM1a$_{1-28}$Zip/Lmod2s1 molar ratios, respectively (S4 Data).** x-Axis shows the Lmod2s1 residue number. Peak intensities in the absence of αTM1a$_{1-28}$Zip (I$_0$) were considered equal to 1. Lmod, leiomodin.
(TIF)

**S8 Fig. Compound chemical shift changes Δδ for $^{15}$N-Lmod2s1 backbone amide $^{15}$N and $^1$H resonances upon binding αTM1a$_{1-14}$Zip.** The compound chemical shift changes (S5 Data) were calculated as $\Delta\delta = [(\Delta\delta_H)^2 + (\Delta\delta_N/6.5)^2]^{1/2}$ [79]. Lmod, leiomodin.
(TIF)

**S9 Fig. Effect of L25G mutation on the binding of Lmod2s1 to αTM1a$_{1-14}$Zip measured by circular dichroism spectroscopy (S6 Data).** Peptide concentrations were 10 μM in 10 mM sodium phosphate buffer (pH 7.0) and 100 mM NaCl. The melting curve of the mixture of Lmod2s1 with αTM1a$_{1-14}$Zip is different from the arithmetic sum of the individual melting curves of free Lmod2s1 and αTM1a$_{1-14}$Zip, indicating that there is interaction between the two peptides. Melting temperature is higher for the mixture, which is consistent with a binding-induced increase in overall α-helical content in the complex [26, 27]. The melting curve of

Lmod2s1[L25G] and $\alpha$TM1a$_{1-14}$Zip mixture is practically identical to the arithmetic sum of the individual curves of free Lmod2s1[L25G] and $\alpha$TM1a$_{1-14}$Zip, indicating no interaction under these conditions. ●, Lmod2s1; ▼, $\alpha$TM1a$_{1-14}$Zip; ■, Lmod2s1 and $\alpha$TM1a$_{1-14}$Zip mixture; □, arithmetic sum of melting curves for Lmod2s1 and $\alpha$TM1a$_{1-14}$Zip; Lmod, leiomodin. (TIF)

**S10 Fig. Superposition of complexes between $\alpha$TM1a$_{1-14}$Zip and TpmBS1 fragments from human Lmod2 (Lmod2s1, tan) and Tmod1 (blue).** Lmod2s1 sequence in the Lmod2s1/$\alpha$TM1a$_{1-14}$Zip complex (Fig 1) was substituted with the homologous sequence from Tmod1 (Tmod1s1), and the Tmod1s1/$\alpha$TM1a$_{1-14}$Zip complex was subjected to a 400-ns MD simulation. A minimal-energy structure (achieved after about 317 ns) was used for comparison. Lmod, leiomodin; MD, molecular dynamics; Tmod, tropomodulin; TpmBS1, tropomyosin-binding site. (TIF)

**S11 Fig. Secondary structure elements in the regions of Lmod2 for which 3D structure is unknown.** (A) Secondary structure elements in the linkers as predicted by Amber18. The fragment Pro42-Thr59 to connect TpmBS1 and ABS1h is colored in orange. The fragment Gly87-Asn195 to connect ABS1h and ABS2 is colored in brown. The secondary structure elements ($\alpha$-helices, shown as coiled ribbons) were formed after 50 ns of MDS. Residues that did not form a regular secondary structure were considered disordered/flexible. The contour lengths of the linkers including the modeled $\alpha$-helices were estimated as 59.5 Å (Pro42-Thr59) and 306 Å (Gly87-Asn195) and were sufficiently large to connect TpmBS1, ABS1h, and ABS2 at the pointed end without steric clashes. (B) Comparison of secondary structure predictions made by Amber18, Jpred4, PsiPred, and PredictProtein for linkers Pro42-Thr59 and Gly87-Asn195. Predicted $\alpha$-helices are highlighted in green. 3D, three-dimensional; ABS, actin-binding site; Lmod, leiomodin; MDS, molecular dynamics simulation; TpmBS1, tropomyosin-binding site. (TIF)

**S12 Fig. Nonuniform broadening of cross-peaks in 2D $^{15}$N-HSQC spectra of $^{15}$N/$^{13}$C-labeled $\alpha$TM1a$_{1-14}$Zip upon addition of unlabeled Lmod2s1.** The spectra were recorded in 50 mM sodium phosphate buffer (pH 6.5), 10% D$_2$O, 0.2 mM EDTA, 0.1% sodium azide, 2X Pierce EDTA-free protease inhibitor cocktail. (A) and (B) are the spectra of $^{15}$N/$^{13}$C-labeled $\alpha$TM1a$_{1-14}$Zip (0.45 mM) in the absence and presence of the excess Lmod2s1 (1.5 molar ratio), respectively. Before the Lmod2s1 addition (spectrum A), 30 resolved cross-peaks corresponding to the backbone amides of $\alpha$TM1a$_{1-14}$Zip (MW $\approx$ 8.2 kDa) were detected. Upon the Lmod2s1 (MW $\approx$ 4.7 kDa) addition (spectrum B), the cross-peaks exhibited non-uniform broadening. As a result of the broadening, the peak intensities decreased, and the total number of the detectable resolved peaks reduced to 23. Additionally, some of the detectable peaks were disproportionally weak (see, e.g., peaks 2, 19, 24, and 26 in spectrum B). The non-uniform broadening suggested that the complex undergoes chemical exchange. The spectra (A) and (B) were recorded at 600 MHz, 10°C. (C) displays an overlay of 2D $^{15}$N-HSQC spectra regions of $^{15}$N/$^{13}$C-labeled $\alpha$TM1a$_{1-14}$Zip in the presence of three different molar ratio of Lmod2s1: $\alpha$TM1a$_{1-14}$Zip, i.e., 1.5 (black), 2.4 (red), and 3 (blue). As the Lmod2s1 concentration increased, we observed an increase of some peak intensities (e.g., cross-peaks from K5 and D14 amides), confirming that the processes of association/dissociation of the complex contribute to the line broadening. The spectra (C) were recorded at 500 MHz, 25°C. 2D, two-dimensional; HSQC, heteronuclear single-quantum coherence; Lmod, leiomodin; MW, molecular mass. (TIF)

**S13 Fig. Chemical exchange in the Lmod2s1/αTM1a$_{1-14}$Zip complex causes non-uniform line broadening in 3D $^{13}$C-correlated spectra, thus imposing limitations on chemical shift assignment.** (A) A 2D $^{15}$N-HSQC spectrum of $^{15}$N/$^{13}$C-labeled αTM1a$_{1-14}$Zip (0.45 mM) in the presence of 1.35 mM Lmod2s1 ([Lmod2s1]:[αTM1a$_{1-14}$Zip] = 3:1). (B) A 2D [$^{15}$N, $^{1}$H]-plane projection of a 3D HNCACB spectrum recorded for the same sample as used in (A). To observe low-intensity peaks, the base contour level was lowered to slightly above the level of noise. Nevertheless, some of the cross-peaks from backbone amides seen in (A) are not present in (B), while many other peaks are weak. Boxed are peaks corresponding to NH strips in panel (C). (C) Strip view of the HNCACB spectrum. Shown in the panel are backbone amide NH strips corresponding to peaks boxed in (B). To observe low-intensity peaks, in strips with weak cross-peaks the base contour level was lowered to the level of noise. Positive and negative peaks are represented by black and red contours, respectively. In many amide NH strips, cross-peaks from Cα(i-1)/Cβ(i-1) of the preceding residue cannot be detected. Notwithstanding the line broadening, a combination of several 3D spectra enabled an almost complete assignment of backbone atoms and partial assignment of side-chain atoms in the complex (see the list of collected spectra in the Materials and Methods). The 2D $^{15}$N-HSQC and 3D HNCACB spectra were recorded in 50 mM sodium phosphate buffer (pH 6.5), 10% D$_2$O, 0.2 mM EDTA, 0.1% sodium azide, 2X Pierce EDTA-free protease inhibitor cocktail, at 500 MHz, 25˚C. 2D, two-dimensional; 3D, three-dimensional; HSQC, heteronuclear single-quantum coherence; Lmod, leiomodin.
(TIF)

**S14 Fig. Examples of peak broadening in $^{13}$C-correlated spectra of $^{15}$N/$^{13}$C-labeled Lmod2s1 in the presence of excess αTM1a$_{1-14}$Zip.** (A) Non-overlapped CH strips of Leu25 (CαH), Leu33 (CαH), Glu18 (CαH), Glu18 (CβH). All Leu25 and Leu33 cross-peaks and some E18 cross-peaks are broadened beyond detection. (B) Shown for comparison are non-overlapped CH strips of I40 (CαH, CβH, Cγ$_2$H, Cδ$_1$H) with sharp cross-peaks and clearly identifiable cross-peak patterns of an Ile side chain. (C) A CH projection of a $^{13}$C-filtered NOESY. Only a few cross-peaks are observed in the spectrum. Positive and negative peaks are represented by black and red contours, respectively. Lmod, leiomodin.
(TIF)

**S15 Fig. $^{1}$H-$^{15}$N RDC values measured for $^{15}$N-Lmod2s1 in the presence of excess αTM1a$_{1-14}$Zip in 5% oriented C8E5/octanol bicelle media.** Uncertainties in RDC determination are shown as vertical bars. The dependence of RDC values on residue number for rigid Glu18-Ser24 and Ala27-Glu38 α-helices of Lmod2s1 follows oscillatory patterns as predicted for regular secondary structure elements (S7 Data) [81, 82]. The fitting of the observed RDCs to a pattern expected for an ideal α-helix gave each helix a set of parameters A$^{I}$, A$^{II}$, and Aυ, which depend on the orientation of the helices with respect to a coordinate system bound to the molecule [81]. A$^{II}$ and Aυ represent dominant sinusoid amplitude and an average RDC value, respectively, for an ideal α-helix [81], and they were found to be different for each of the two helices (A$^{II}$~5.2 ± 0.2 Hz, Aυ = 11.0 ± 0.1 Hz for Glu18-Ser24 and A$^{II}$ = 2.7 ± 0.2 Hz, Aυ = 12.2 ± 0.1 Hz for Ala27-Glu38). Specifically, the amplitude A$^{II}$ was larger for helix Glu18-Ser24, and Aυ was slightly larger for residues Ala27-Glu38. According to the theoretical expressions for A$^{II}$ and Aυ [81], the helices are angled with respect to each other in the complex, in agreement with the result obtained by TALOS+. Lmod, leiomodin; RDC, residual dipolar coupling.
(TIF)

**S16 Fig. Correlation between measured $^{1}$H-$^{15}$N RDC values and those calculated by RED-CAT for residues Glu18-Glu38 of Lmod2s1 (S8 Data).** Vertical bars show estimated

uncertainties in RDC values. RDC, residual dipolar coupling.
(TIF)

**S1 Table. Outputs of TALOS+ for Lmod2s1 and αTM1a$_{1\text{-}14}$Zip.** PHI and PSI are predicted *i*th residue's torsion angles φ (atoms C'$_{i-1}$-N$_i$-Cα$_i$-C'$_i$) and ψ (atoms N$_i$-Cα$_i$-C'$_i$-N$_{i+1}$), DPHI and DPSI are the estimated standard deviations, DIST is the TALOS+ database matching score, S2 is the Wishart chemical shift order parameter RCI-$S^2$ [83], COUNT is the number of database triplets used to form the torsion angle predictions, CLASS is the classification of the predicted results, where "None" corresponds to "no torsion prediction was made," "Good" corresponds to majority consensus in database matches, "Warn" corresponds to no consensus in database matches, and "Dyn" corresponds to an RCI-$S^2$ value that indicates that the residue has dynamic conformation. RCI-$S^2$ is typically >0.85 for a rigid residue, and it reduces as the residue experiences more internal motions [60, 83]. The data for each peptide are stored in a separate worksheet tab. Lmod, leiomodin; RCI, random coil index.
(XLSX)

**S2 Table. Pairwise RMSD values (in angstroms) between Lmod2s1/αTM1a$_{1\text{-}14}$Zip structures after simulated annealing.** Each simulated annealing structure was produced from one of two starting structures, "model 1" or "model 2." "Model 1" was a structure with the long (Ala27-Glu38) Lmod2s1 helix parallel to the tropomyosin coiled coil, whereas "model 2" was a structure with the short (Glu18-Ser24) Lmod2s1 helix parallel to the tropomyosin coiled coil. Five replicas (#1–#5) were analyzed for each starting point. To calculate RMSD, backbone atoms N, Cα, and C' of well-structured residues Asp17-Asp39 (Lmod2s1) and Asp2-Leu28 (αTM1a$_{1\text{-}14}$Zip) in two coordinate files were first matched by UCSF Chimera. For the matched structures, a global RMSD (between the same backbone atoms, top table) and a RMSD for the Lmod2s1 peptide (between backbone atoms N, Cα, and C' of residues Asp17-Asp39 in Lmod2s1, bottom table) were calculated. Lmod, leiomodin; RMSD, root-mean-square deviation.
(XLSX)

**S3 Table. RMSD (in angstroms) for pairs of 10 Lmod2s1/αTM1a$_{1\text{-}14}$Zip MD replicas (PDB ID 6UT2).** The RMSD were calculated using coordinates of superimposed backbone atoms N, Cα, and C' of well-defined regions in the molecular complex (Asp17-Asp39 in Lmod2s1 and Asp2-Leu28 in αTM1a$_{1\text{-}14}$Zip). The RMSD sample mean and SEM were calculated at 1.76 Å and 0.07 Å, respectively. Lmod, leiomodin; MD, molecular dynamics; PDB, Protein Data Bank; RMSD, root-mean-square deviation.
(XLSX)

**S1 Video. An 80-ns MD movie for a left-hand parallel packing of the Lmod2s1/αTM1a$_{1\text{-}14}$Zip complex.** In the simulation, the temperature was set to 300 K, volume to constant, and the Langevin thermostat collision frequency was 1 ps$^{-1}$. Lmod, leiomodin; MD, molecular dynamics.
(MP4)

**S2 Video. An 80-ns MD movie for a left-hand antiparallel packing of the Lmod2s1/αTM1a$_{1\text{-}14}$Zip complex.** In the simulation, the temperature was set to 300 K, volume to constant, and the Langevin thermostat collision frequency was 1 ps$^{-1}$. Lmod, leiomodin; MD, molecular dynamics.
(MP4)

**S3 Video. An 80-ns MD movie for a right-hand parallel packing of the Lmod2s1/αTM1a$_{1\text{-}14}$Zip complex.** In the simulation, the temperature was set to 300 K, volume to constant, and

the Langevin thermostat collision frequency was 1 ps$^{-1}$. Lmod, leiomodin; MD, molecular dynamics.
(MP4)

**S4 Video. An 80-ns MD movie for a right-hand antiparallel packing of the Lmod2s1/αTM1a$_{1-14}$Zip complex.** In the simulation, the temperature was set to 300 K, volume to constant, and the Langevin thermostat collision frequency was 1 ps$^{-1}$. Lmod, leiomodin; MD, molecular dynamics.
(MP4)

**S5 Video. An 80-ns MD movie for a crisscross packing of the Lmod2s1/αTM1a$_{1-14}$Zip complex.** In the simulation, the temperature was set to 300 K, volume to constant, and the Langevin thermostat collision frequency was 1 ps$^{-1}$. Lmod, leiomodin; MD, molecular dynamics.
(MP4)

**S1 Data. Thin filament length measurements in rat cardiomyocytes transfected with either GFP, GFP-Lmod2 WT, or GFP-Lmod2[L25G].** Excel spreadsheet containing raw data for Fig 4B. Description of what is measured along with the units can be found in the first cell of the first column. GFP, green fluorescent protein; Lmod, leiomodin; WT, wild type.
(XLSX)

**S2 Data. Percentage of rat cardiomyocytes transfected with either GFP, GFP-Lmod2 WT, or GFP-Lmod2[L25G] showing consistent Tmod1 assembly.** Excel spreadsheet containing raw data for Fig 5B. Description of what is measured along with the units can be found in the first cell of the first column. GFP, green fluorescent protein; Lmod, leiomodin; Tmod, tropomodulin; WT, wild type.
(XLSX)

**S3 Data. A pdb file with the model displayed in Fig 7.** Since no experimental structural information is available for residues Pro42-Thr59 and Gly87-Asn195, their true conformation and atomic coordinates can deviate significantly from the model.
(PDB)

**S4 Data. Titration of $^{15}$N-Lmod2s1 with αTM1a$_{1-28}$Zip.** Excel spreadsheet containing raw data for S7 Fig. Lmod, leiomodin.
(XLSX)

**S5 Data. Compound chemical shift changes Δδ for $^{15}$N-Lmod2s1 backbone amide $^{15}$N and $^{1}$H resonances upon binding αTM1a$_{1-14}$Zip.** Excel spreadsheet containing raw data for S8 Fig. Lmod, leiomodin.
(XLSX)

**S6 Data. Thermally induced unfolding of Lmod2s1 WT, Lmod2s1 [L25G], and αTM1a$_{1-14}$Zip peptides and their complexes tracked by circular dichroism at 222 nm.** Excel spreadsheet containing the raw and processed data for production of S9 Fig. Descriptions and units of each column of data are located in the first cell of the column. Lmod, leiomodin; WT, wild type.
(XLSX)

**S7 Data. $^{1}$H-$^{15}$N RDC values measured for $^{15}$N-Lmod2s1 in the presence of excess αTM1a$_{1-14}$Zip in 5% oriented C8E5/octanol bicelle media.** Excel spreadsheet containing raw data for S15 Fig. Lmod, leiomodin; RDC, residual dipolar coupling.
(XLSX)

**S8 Data. Correlation between measured and calculated RDC values.** Excel spreadsheet containing raw data for S16 Fig. RDC, residual dipolar coupling.
(XLSX)

## Acknowledgments

A portion of the NMR data collection was performed using EMSL (grid.436923.9), a DOE Office of Science User Facility sponsored by the Office of Biological and Environmental Research. We gratefully acknowledge the support of NVIDIA Corporation for the donation of the Titan Xp GPU used in this study. We thank Rachel Mayfield and Tania Larrinaga for technical assistance with neonatal rat cardiomyocyte cultures and Dr. Vitold Galkin for discussing the pointed-end model.

## Author Contributions

**Conceptualization:** Dmitri Tolkatchev, Carol C. Gregorio, Alla S. Kostyukova.

**Data curation:** Dmitri Tolkatchev, Garry E. Smith, Jr., Lauren E. Schultz, Mert Colpan.

**Formal analysis:** Dmitri Tolkatchev, Garry E. Smith, Jr., Lauren E. Schultz, Mert Colpan, Alla S. Kostyukova.

**Funding acquisition:** Carol C. Gregorio, Alla S. Kostyukova.

**Investigation:** Dmitri Tolkatchev, Garry E. Smith, Jr., Lauren E. Schultz, Mert Colpan, Gregory L. Helms, John R. Cort, Alla S. Kostyukova.

**Methodology:** Dmitri Tolkatchev, Garry E. Smith, Jr.

**Project administration:** Carol C. Gregorio, Alla S. Kostyukova.

**Resources:** Dmitri Tolkatchev, Garry E. Smith, Jr., Lauren E. Schultz, Mert Colpan, Gregory L. Helms, John R. Cort, Carol C. Gregorio, Alla S. Kostyukova.

**Supervision:** Dmitri Tolkatchev, Carol C. Gregorio, Alla S. Kostyukova.

**Validation:** Dmitri Tolkatchev, Garry E. Smith, Jr., Lauren E. Schultz, Carol C. Gregorio.

**Visualization:** Dmitri Tolkatchev, Garry E. Smith, Jr., Lauren E. Schultz, Mert Colpan.

**Writing – original draft:** Dmitri Tolkatchev, Lauren E. Schultz, Mert Colpan, Carol C. Gregorio, Alla S. Kostyukova.

**Writing – review & editing:** Dmitri Tolkatchev, Garry E. Smith, Jr., Lauren E. Schultz, Mert Colpan, Gregory L. Helms, John R. Cort, Carol C. Gregorio, Alla S. Kostyukova.

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
