## [Editor Report · Decision Letter 0]

20 Mar 2020

Dear Dr Tolkatchev, 

Thank you for submitting your manuscript entitled "Leiomodin creates a “leaky” cap at the pointed end of thin filaments" for consideration as a Research Article by PLOS Biology.

Your manuscript has now been evaluated by the PLOS Biology editorial staff as well as by an academic editor with relevant expertise and I am writing to let you know that we would like to send your submission out for external peer review.

Please re-submit your manuscript within two working days, i.e. by Mar 22 2020 11:59PM.

Kind regards,

Di Jiang,

Associate Editor

PLOS Biology

---

## [Decision Letter · Decision Letter 1]

29 Apr 2020

Dear Dr Tolkatchev,

Thank you very much for submitting your manuscript "Leiomodin creates a “leaky” cap at the pointed end of thin filaments" for consideration as a Research Article at PLOS Biology. Your manuscript has been evaluated by the PLOS Biology editors, an Academic Editor with relevant expertise, and by four independent reviewers.

In light of the reviews (below), we will welcome re-submission of a revised version that takes into account the reviewers' comments. We cannot make any decision about publication until we have seen the revised manuscript and your response to the reviewers' comments. Your revised manuscript is also likely to be sent for further evaluation by the reviewers.

We expect to receive your revised manuscript within 2 months. Please note given we are in the midst of COVID-19, we are flexible regarding turnaround time for revision. 

**IMPORTANT - SUBMITTING YOUR REVISION**

*Re-submission Checklist*

*Published Peer Review*

*PLOS Data Policy*

*Blot and Gel Data Policy*

Sincerely,

Di Jiang, PhD

Associate Editor

PLOS Biology

REVIEWS:

Reviewer #1: This manuscript combines structural data (NMR & MD simulations) and experiments based on this to have a closer look at the mechanism of leiomodin action a the pointed end of thin filaments. The structural data demonstrates why leiomodin is restricted to the pointed end and does not associate further along the thin filament and pinpoints Leu25 in the Lmods2s1 fragment as the crucial residue to interact with N-terminal tropomyosin, which would along the thin filament be obscured by the overlap with the next tropomyosin. Mutation of this amino acid to a glycine reduced the association of leiomodin with the pointed ends in cardiomyocytes, prevented the extension of thin filaments that would be seen with wild type leiomodin over-expression and maintained the targeting of tropomodulin, which would get replaced by over-expressed leiomodin. The authors subsequently present a really nice model how leiomodin on the pointed end of a thin filament might work by acting as a swinging gate to allow association of actin monomers.

These data are extremely interesting, since they help to explain hitherto puzzling differential behavior of pointed end thin filament-associated proteins. The experimental evidence is clear and convincing and my major comments are on presentation issues:

Figure 7: It would be easier to interpret this figure, if actin monomers would be presented in two different shades of color? ABS2 is shown in magenta not purple? Its structure is modelled from the tropomodulin structure? Clearly state this in legend and text.

Figure 8: turquoise new actin attaches to actin A not B as stated?

Reviewer #2: This is a very nice article addressing a very important question in muscle biology which relates to the precise mode of interaction of leimodulin (Lmod) with tropomyosin (tpm) and actin at the pointed ends of thin filaments and how leimodulin and tropomodulin (Tmod) compete for this site leading to very differenent outcomes regarding filament growth. So far there were structures available for some fragments of Lmod with actin but the crucial interaction with tpm had not been described. This manuscript finally closes the gap in a comprehensive way. Structure determination of a complex of small fragments of Lmod and tpm is performed using some NMR data which are used to inform and validate computer simulations that are used to generate the complex. The model structure of the complex is then explode in cardiomyocytes using a mutant (L25G) which abolishes the Lmod-tpm interaction. The structure is then used to develop a useful model to explain the function of Lmod at the pointed end and how its interaction at this site diverges from that of Tmod.

While the overall arguments are in general convincing, the NMR experiments appear extremely limited and the structure determination relying too much on model building. The main recommendation from this reviewer would therefore be to strengthen the experimental data to provide a better underpinning for the modelling. Furthermore, there should be more quantitative information about the quality of the structure. 

In detail I recommend the following:

1. The authors mention that the assignment is incomplete due to resonance overlap. However, a broad range of triple resonance experiments were measured yet one of the most important ones that helps in this case - of the type of HN(CO)CCH or HNCCH - were not recorded. Given the size of the complex and the relatively high concentration this should be feasible. These experiments provide all side chain proton and carbon resonances at the resolution of the HSQC which is excellent. A more complete assignment will also be useful in better ways of model building.

2. The authors indirectly create order parameters from chemical shifts. However, it would be much better to explicitly measure 15N relaxation data (heteronuclear NOE, T1, T2) which would give precise dynamics information residue by residue. Such data could also be used in the validation of the structure by considering anisotropic rotational diffusion of the complex.

3. It is most unfortunate the RDS could only be measured for one of the two peptides in the complex. Oriented media made from lipids or organic solvents are often challenging for certain proteins and it is not surprising that they caused problems in this case. Alternatives such as purple membrane or pf1 phage which are much more inert should be considered.

4. In addition to using the RDCs there should be more precise measures of the consistency of the structure analogous to resolution in X-ray or RMSD in traditional NMR based structures. This will allow the reader a better assessment of the robustness of the structure. 

5. The only use that the authors make from their assignments in structure calculation is the creation of dihedral angle constraints in TALOS+. It would be useful to consider alternative approaches that allow a much more comprehensive use of full assignments such as that provided by Rosetta-CS which is able to use the entire assignment and not only the backbone atoms and thus offers a much better representation of the experimental data. Additionally, by creating a structural model from protein fragments it makes for a less biased approach for model building.

Reviewer #3: The paper by Tolkatchev et al. proposes a model for the molecular mechanism of actin polymerisation at the pointed end in the presence of leiomodin based on NMR experiments and MD simulations. 

The manuscript is well written and the results potentially interesting. However, I feel that some points need further clarification:

* more details should be provided for the MD simulations performed on Lmod2s1/αTM1a1-14Zip. In particular:

- shape of the box

- cutoff values used for non bonded interactions

- units of the force constants used for the dihedral restraints

- type of barostat with coupling constant

* MD simulations were performed also on the linkers connecting the Lmod2 binding sites, but no methodological details were provided. The Authors need to indicate the simulation parameters and the protocol followed for these systems. Also, it would be useful to know if the secondary structure from MD is consistent with secondary structure predictions (give for example by PSIPRED or analogous bioinformatics software). Additionally, it is unclear where these linkers are in Figure 7.

* More quantitative data should be provided for:

- the preliminary runs on the alternative packing of the helices, showing the instability of the side-by-side and stability of the criss-cross arrangements in all the replicas.

- the results of the Simulated Annealing simulations: it is said that "The annealed structures were very similar", but this needs to be shown (for example with a superimposition of the structures or calculation of RMSD values).

* at page 13, lines 355-356: "confirming that the N-terminus of tropomyosin

cannot accommodate a simultaneous binding of both Tmod1 and Lmod2" is too strong considering how this result was obtained (simple replacement of the sequence followed by a relatively short MD simulation). The statement should be revised to highlight that this is a model.

* the final structures of the models should be made available as supplementary data. 

Reviewer #4 (Frank Schnorrer, signed review): This manuscript by Tolkatchev and colleagues investigates the molecular structure how the actin capping protein leiomodin (Lmod2) interacts with tropomyosin (TM) at the pointed ends of actin filaments and tests predictions in a cardiomyocyte model in culture. 

This is relevant as the current literature discusses 2 distinct models how Lmod2 and its close homolog Tropomodulin (Tmod) regulate thin filament length in sarcomeres. In the 'competition model' they displace each other from the pointed ends and thus excess Lmod2 can elongate, whereas excess Tmod can cap actin filament pointed ends preventing further polymerisation. An alternative model suggests that Lmod2 solely acts as a thin filament nucleator of new filaments.

The authors use NMR to determine the structure and apply molecular dynamics simulations of the relevant Lmod2 and TM peptides to develop a detailed molecular model of the interaction of the N-term of Lmod2 with the N-term of the TM dimer.

This model enables experimental predictions how to block the interaction of Lmod2 with TM by a L25G mutation in the Lmod2 peptide. Indeed, transfection of cardiomyocytes shows that the L25G mutant Lmod protein cannot localise to the sarcomeric M-band and cannot increase thin filament length, compared with a wild-type construct. It can also not displace Tmod from the pointed ends sarcomeric actin filaments.

Together, this allowed the authors to build a detailed molecular model how Lmod2 can promote pointed end actin polymerisation suggesting a 'swinging gate' model of actin binding at the pointed end. This model clearly supports the competition model of Lmod2 versus Tmod binding with only one of the two binding at a given time to the TM dimer. I also provides an explanation why Lmod2 and Tmod are enriched at the final TM dimer at the pointed actin filament ends.

Overall, I find the paper nicely written and convincing for a cell and developmental biologist with interesting implications towards the mechanism of myofibrils and sarcomere formation during development. 

I am not a structural biologist and cannot judge the methodology of the NMR and the molecular simulations. However, the results are well illustrated and make sense to me. The experimental validations in the cardiomyocyte model fully support the structure and the predictions of the molecular model.

Minor points.

1. I am not a fan of the 'leaky' cap expression present in title (but not in the abstract). Would not 'dynamic' cap be better? As opposed to 'stable' cap for Tmod. In any case, if this expression is important enough to make it into the title it should also appear in the abstract when the molecular model is discussed.

2. Figure 4 should include the GFP-Lmod2 channel separately in grey to better illustrate the localisation of the wild type and mutant proteins. Does the sarcomere length change upon GFP-Lmod2 transfection?

3. Of course, it would be nice to engineer a cardiomyocyte cell line with CRISPR that contains the L25G in Lmod to demonstrate the functional relevance, at least in the in vitro model. However, I can see this experiment being challenging in the current situation with limited access to the labs. Hence, I would not consider it essential.

---

## [Decision Letter · Decision Letter 2]

13 Jul 2020

Dear Dr Tolkatchev,

Thank you for submitting your revised Research Article entitled "Leiomodin creates a “leaky” cap at the pointed end of thin filaments" for publication in PLOS Biology. I have now obtained advice from the original reviewers and have discussed their comments with the Academic Editor. 

Based on the reviews, we will probably accept this manuscript for publication, assuming that you will modify the manuscript to address the remaining points raised by the reviewers. We also strongly encourage you to edit the Title of the paper from "Leiomodin creates a “leaky” cap at the pointed end of thin filaments" to "Leiomodin creates a leaky cap at the pointed end of actin thin filaments" (removing the quotation marks and adding the word "actin"). In addition, because your study used neonatal cardiomyocytes isolated from rats, which are vertebrate tissues, you will need to add an Ethics Statement (please see below). 

Please also make sure to address the data and other policy-related requests noted at the end of this email.

We expect to receive your revised manuscript within two weeks. Your revisions should address the specific points made by each reviewer. In addition to the remaining revisions and before we will be able to formally accept your manuscript and consider it "in press", we also need to ensure that your article conforms to our guidelines. A member of our team will be in touch shortly with a set of requests. As we can't proceed until these requirements are met, your swift response will help prevent delays to publication.

*Copyediting*

*Published Peer Review History*

*Early Version*

*Submitting Your Revision*

Sincerely,

Di Jiang, PhD 

Senior Editor

PLOS Biology

ETHICS STATEMENT:

-- Please create a separate Ethics Statement and place it in the beginning of the Methods section. Please include all relevant information described below including an approval number. 

-- Please include the full name of the IACUC/ethics committee that reviewed and approved the animal care and use protocol/permit/project license. Please also include an approval number.

-- Please include the specific national or international regulations/guidelines to which your animal care and use protocol adhered. Please note that institutional or accreditation organization guidelines (such as AAALAC) do not meet this requirement.

-- Please include information about the form of consent (written/oral) given for research involving human participants. All research involving human participants must have been approved by the authors' Institutional Review Board (IRB) or an equivalent committee, and all clinical investigation must have been conducted according to the principles expressed in the Declaration of Helsinki.

DATA POLICY:

Regardless of the method selected, please ensure that you provide the individual numerical values that underlie the summary data displayed in the following figure panels as they are essential for readers to assess your analysis and to reproduce it: Figs 4B, 5B, S7, S8, S9, S12, S13. NOTE: the numerical data provided should include all replicates AND the way in which the plotted mean and errors were derived (it should not present only the mean/average values).

Reviewer remarks:

Reviewer #1: My comments were all addressed in the revised version.

Reviewer #2: The original draft of the manuscript didn't mention the complex dynamics of the complex and the challenges resulting from that. While it is still disappointing that certain experiments were not attempted (e.g. one can get around the exchange contribution to 15N T2 by measuring the transversal cross correlation relaxation rate) the authors have addressed all issues raised in a reasonable manner. The changes made to the manuscript make it now a bit more clear what the experimental constraints were and why certain data are not available. 

The only minor change that I would request is that the authors should add to the supplementary material some of the already recorded spectra that are mentioned in the response to the referee questions. E.g. it is mentioned that some line width reduction was observed in 15N HSQC spectra as a function of concentration of unlabelled peptide. It is also mentioned that some 13C correlated 3D spectra and some X-filtered NOESY spectra had only broadened peaks. It would be very useful if the 2D spectra and selected slices from the 3D spectra would be made available so that the interested reader could get a better idea of the dynamics of the system.

Reviewer #3: The authors have addressed all my comments. About the release of the pdb structure of the complete model, I still think it needs to be made available as supplementary material, you can just state clearly that it is a model to avoid any confusion.

Reviewer #4: As detailed in my initial review I find this paper nicely written and convincing for a cell and developmental biologist with interesting implications towards the mechanism of myofibril and sarcomere formation during muscle development. My view has been strengthened after re-reading the revised manuscript.

As I also pointed out I am not a structural biologist and cannot contribute to the methods discussion of the NMR and the molecular simulations, which have been the major points during this revision. The minor points that I had raised in my review have all been addressed and the results support the main hypothesis of the authors. Hence, I would support publication.

Comment:

The authors had uploaded Figure 4 twice and Figure 5 was missing from the revised version, but there was no change requested on Figure 5.

---

## [Editor Report · Decision Letter 3]

17 Aug 2020

Dear Dr Tolkatchev,

On behalf of my colleagues and the Academic Editor, Simon M Hughes, I am pleased to inform you that we will be delighted to publish your Research Article in PLOS Biology. 

Early Version

PRESS 

Kind regards,

Alice Musson

Publishing Editor, 

PLOS Biology

on behalf of

Ines Alvarez-Garcia,

Senior Editor

PLOS Biology